# The limits to growth of northern peatland carbon stocks

Georgii A. Alexandrov[1], Victor A. Brovkin[2], Thomas Kleinen[2], Zicheng Yu[3,4]

[1] A.M. Obukhov Institute of Atmospheric Physics, Russian Academy of Sciences, Pyzhevsky 3, Moscow, 119017, Russia
[2] Max Planck Institute for Meteorology, Bundesstrasse 53, 20146 Hamburg, Germany
[3] Department of Earth and Environmental Sciences, Lehigh University, 1 West Packer Avenue, Bethlehem, PA 18015, USA
[4] Institute for Peat and Mire Research, School of Geographical Sciences, Northeast Normal University, Changchun 130024, China

*Correspondence to*: Victor A Brovkin (victor.brovkin@mpimet.mpg.de)

**Abstract.** Northern peatlands have been a persistent natural carbon sink since the last glacial maximum. If there were no limits to their growth, carbon accumulation in these ecosystems could offset a large portion of anthropogenic carbon emissions until the end of the present interglacial period. The limits to the growth of northern peatland carbon stocks, evaluated based on the gridded data on the depth to bedrock and on the fraction of area covered by soils of histosol type, suggest that 875±125 PgC is the most expedient estimate of the potential carbon stock in northern peatlands at large and that 330±200 PgC is the most expedient estimate of the total amount of carbon that they could remove from the atmosphere during the period from present to the end of the current interglacial. This leads to conclusion that northern peatlands, not only the oceans, will potentially play an important role in reducing the atmospheric carbon dioxide concentration over the next five thousand years.

## 1 Introduction

The recent compilations of peatland data (Loisel et al., 2014; Treat et al., 2019) largely confirm the conventional notion of the carbon (C) sink provided by northern peatlands, namely the peatlands distributed across the northern mid- and high-latitude regions located north of 45°N, since the Last Glacial Maximum (Loisel et al., 2017). According to this notion, northern peatlands were providing a persistent but variable sink for atmospheric carbon (Yu, 2011). Variations in the sink magnitude are explained by changes in the rate of peatland expansion and in the rate of peat accumulation. In the early Holocene, both the rate of peatland expansion and the rate of carbon accumulation appear to be highest (Yu et al., 2010) as compared to the later Holocene periods. Since the area of peatlands remained relatively stable in the late Holocene (Adams and Faure, 1998; MacDonald et al., 2006; Yu et al., 2010), the major part of the carbon sink provided by northern peatlands during this period could be attributed to the growth in peat depth, not to the growth of the area occupied by the northern peatlands.

The average rate of carbon accumulation associated with peat growth is estimated at 18-28 gC $m^{-2}$ $yr^{-1}$ (Yu, 2011). This rate suggests that northern peatlands, occupying 2.4-4 million $km^2$ (Yu, 2011), may accumulate during the next 20,000 years the amount of carbon comparable to the expected cumulative anthropogenic carbon emissions corresponding to a 2.5°C warming (Raupach et al., 2014), namely from 864 (18 gC $m^{-2}$ $yr^{-1}$ ×2.4·$10^{12}$ $m^2$ × 2·$10^4$ yr) to 2240 (28 gC $m^{-2}$ $yr^{-1}$ × 4·$10^{12}$ $m^2$ × 2·$10^4$ yr) PgC.

There has been little research, however, on estimating the potential magnitude of the cumulative carbon removal associated with the natural development of peatland ecosystems. Individual peatland development may lead to reduction of the carbon sink potential: the closer the peatland ecosystem is to its steady state, that is, to the equilibrium between organic matter production and decomposition, the lower is the carbon sink magnitude. Therefore, the amount of carbon that northern peatlands could remove from the atmosphere will be less than that estimated above.

The process of reaching equilibrium can be conceptualized as follows, see also (Clymo, 1984; Alexandrov et., 2016). Peat is accumulated due to protection of plant litters in the catotelm, the lower layer of a peat deposit that is permanently saturated with water. The plant litters do not enter the catotelm directly, but instead they first enter the upper layer of the peat deposit, the acrotelm, that is not permanently saturated with water. Despite intense aerobic decomposition of organic matter in the acrotelm, at least a small portion of the organic matter that enters the acrotelm always reaches the catotelm in an accumulating peatland. This is, of course, not true in the case of a degrading peatland, but degrading peatlands do not fall within the scope of this study.

Precisely speaking, the organic matter does not reach the catotelm, it is rather "flooded" by elevating groundwater. The rise of groundwater is caused by the rise of the peatland surface that in turn results from accumulation of organic matter. This loop cannot elevate the groundwater infinitely. The maximum height of the water table, and thus the potential peat depth, is determined by the amount of effective rainfall, drainage system density (the length of draining streams per unit area) and the hydraulic conductivity of peat and mineral materials below the peat (Alexandrov et al., 2016).

The purpose of our study is to estimate the potential peat depth and carbon stocks over NH area north of 45°C and arrive to conclusion about the cumulative carbon removal associated with the natural development of northern peatlands by the end of the current interglacial. Although it is not completely clear how long the current interglacial will last, the recent attempts to estimate its possible duration lead to conclusion that a glacial inception is unlikely to happen within the next 50,000 years if cumulative carbon emissions exceed 1000 PgC (Berger et al., 2016). Since the duration of the current interglacial depends on the cumulative carbon emissions, it should also depend on the cumulative carbon removal that may offset the effect of carbon emissions, and therefore our study contributes also to the discussion on whether the Earth System would remain in the present delicately balanced interglacial climate state for an unusually long time.

## 2 Methods

### 2.1 Model equations

To calculate the potential peat depth, we apply an equation derived (see Supplement) from the impeded drainage model used in our previous study (Alexandrov et al., 2016). This equation relates the maximum height of the water table above the level of the draining system, $h_{max}$, at a given watershed to the fraction of its area covered by peatland, $f_{P,obs}$, and the average depth to bedrock, $g$:

$$h_{max} = \frac{g}{\sqrt{1 - f_{P,obs}}} \tag{1}$$

This allows us (see Supplement) to estimate, based on gridded data of soil properties (Batjes, 2016) and depth to bedrock (Hengl et al., 2014), the potential average peat depth, $p_{d,max}$, in a grid cell as

$$p_{d,max} = \left( (h_{max} - g) - \frac{1}{3}\left( h_{max} - g\left(\frac{g}{h_{max}}\right)^2 \right) \right) \frac{1}{f_{P,obs}} + d \tag{2}$$

where $d$ is the maximum depth of acrotelm, in m (set at 0.4 m), and then to estimate the maximum carbon stock in the grid cell, $p_{C,max}$ as

$$p_{C,max} = c \times A \times f_{P,obs} \times p_{d,max} \tag{3}$$

where $c$ is the bulk carbon density of peat, in gC m$^{-3}$ (set at 58 KgC m$^{-3}$); $A$ is the area of the grid cell in m$^2$.

### 2.2 Input data

The values of $g$ at the cells of 0.1°×0.1° geographic grid (Figure 1) were estimated from the data on depth to bedrock (Hengl et al., 2014). The use of these data for estimating $g$ in permafrost landscapes is somewhat challenging, because the hydraulic conductivity of permafrost could be as low as that of bedrock under some conditions. Due to this reason, we find it more suitable to use the maximum depth of the active layer for estimating $g$ on these landscapes, for example, by setting $g$ at 2 meters for the regions where mean annual temperature is below -2°C, that is, assuming that the southern boundary of permafrost could be approximated by the -2°C isotherm of mean annual temperature (Riseborough et al., 2008) and that the active layer thickness does not exceed 2 m. The latter is an *ad hoc* assumption based on the recent discussion of uncertainties in the methods for estimating active layer thickness at regional scale (Mishra et al., 2017).

To determine the present-day peatland extent, we relied on the WISE30sec data set (Batjes, 2016) of soil properties at 30'' resolution. The data set contains a classification of soil type for each mapping unit, and to diagnose peatland extent we determined the fraction of each 0.1°×0.1° grid cell covered by soils of histosol type (soil code HS in FAO90 classification). These data allow us to estimate the values that $f_{P,obs}$ may take at the cells of the 0.1°×0.1° geographic grid (Figure 2) and the total area, 2.86 ×10$^6$ km$^2$, that peatlands occupy in the land north of 45ºN.

This estimate of the peatland area does not go beyond the recent estimates (Yu, 2012) (that fall in the range of 2-4 million km$^2$), but it cannot be easily interpreted as the actual peatland area. The estimates of the actual peatland area may vary depending on the criteria that are used to distinguish peatlands from other types of land surface. The minimal depth of the peat layer, which is used to classify a land unit as peatland, is the criterion that affects the estimates of peatland area (Xu et al., 2018). Since peatland extent is diagnosed by the extent of histosols, $2.86 \times 10^6$ km$^2$ should be interpreted as an estimate of the area of peatlands with peat depth exceeding 40 cm (according to FAO definition of histosols).

2.3 Uncertainty associated with peatlands distribution over a grid cell

The gridded data on soil properties (Batjes, 2016) give the fraction of a grid cell covered by peatlands. To estimate the fraction of a watershed covered by peatlands, $f_{PW}$, which is needed for calculating $h_{max}$, one should make an assumption about the peatland distribution within the grid cell. This problem can be illustrated with the following example. The fact that 36% of a grid cell is covered by peatlands ($f_{P,obs}=0.36$) may mean that peatlands cover 36% of each watershed within the grid cell ($f_{PW}=0.36$), or that only 48% of watersheds are occupied by peatlands ($f_{WP}=0.48$), and peatlands cover 75% of each of these watersheds ($f_{PW}=0.75$; $f_{P,obs} = f_{WP} \times f_{PW} =0.48\times0.75=0.36$).

We address this uncertainty by giving three estimates of the potential amount of carbon that could be accumulated in northern peatlands: the uniform estimate, the clumped estimate and the conductivity-dependent estimate. The uniform estimate assumes a uniform distribution of peatlands over all grid cells ($f_{PW}= f_{P,obs}$; $f_{WP}=1$), the clumped estimate assumes a non-uniform distribution over all grid cells ($f_{PW}=0.75$; $f_{WP}=f_{P,obs}/0.75$), and the conductivity-dependent estimate is derived using a rule-based algorithm categorizing the grid cells into those where peatland distribution is uniform and those where peatland distribution is non-uniform. The value of the hydraulic conductivity coefficient, K, calculated from the amount of annual precipitation, potential evapotranspiration, $f_{p.obs}$, and $g$ (see Supplement) is used in this algorithm as an indicator of non-uniform peatland distribution within a grid cell. If K is above the typical value, $K_c$, then it can be assumed that peatland occupy $f_{p.obs}$ / $f_{p.est}$ fraction of watersheds and cover $f_{p.est}$ fraction of area of each of these watersheds, where $f_{p.est}$ is set at the value that brings K to $K_c$.

The typical values of hydraulic conductivity vary in a relatively wide range. Due to this reason, we set $K_c$ at the value that leads the estimate of the potential carbon stocks in northern peatlands to that implied by the peat decomposition model employed by Yu (Yu, 2011). This model suggests that the growth of carbon stock in peatlands is limited by the ratio of annual C input to catotelm to the decay constant. Based on the data from peat cores, the annual C input to catotelm is estimated at 74.8 TgC yr$^{-1}$ and decay constant at 0.0000855 yr$^{-1}$ (Yu, 2011). Thus, the potential carbon stock in northern peatlands could be estimated at 875 PgC (74.8/0.0000855=874,853.8 TgC $\approx$875 PgC), and due to uncertainty in the annual C input to catotelm and decay constant may range from 750 to 1000 PgC (see Supplement). Therefore, we set $K_c$ at the value, namely at 157 m yr$^{-1}$ ($\approx 0.5\times10^{-5}$ m s$^{-1}$), that makes the conductivity-dependent estimate of the potential carbon stocks in northern peatlands equal to 875 PgC.

The use of this approach to addressing uncertainty is illustrated by Table 1, where the estimates of potential peat carbon density in the central part of peatlands are compared to the values observed at 33 peatland sites (Billings, 1987; Borren et al., 2004; Jones et al., 2009; Robinson, 2006; Turunen et al., 2001; Yu et al., 2009). As it can be seen from Table 1, the estimates of the potential peat carbon density based on the uniform interpretation of $f_{P,obs}$ ($f_{PW}= f_{P,obs}$; $f_{WP}=1$) are often lower than the actual peat carbon density at the sites that fall within the cells where $f_{P,obs}$ is low. For example, the actual peat carbon density at site #30, a raised bog that falls within a cell of which 6% are covered by peatlands, is equal to 214 kgC m$^{-2}$, whereas the estimate of the potential peat carbon density based on the uniform interpretation of $f_{P,obs}$ is equal to 65 kgC m$^{-2}$. This example shows that in this case assuming a uniform distribution of peatlands could be wrong. The clumped interpretation of $f_{P,obs}$ ($f_{PW}=0.75$; $f_{WP}=0.08$) gives much higher value of the potential peat carbon density, 1350 kgC m$^{-2}$, that, in its turn, may overestimate the potential peat carbon density at this site if the bog covers less than 75% of the watershed area. The conductivity-dependent interpretation of $f_{P,obs}$ (for $K_c =157$ m yr$^{-1}$ ) suggests that the bog covers 53% of the watershed area and its potential peat carbon density is equal to 636 kgC m$^{-2}$.

## 3 Results

The conductivity-dependent estimates of the potential carbon stocks in the cells of 0.1°×0.1° geographic grid for $K_c =157$ m yr$^{-1}$ are shown on Figure 3 (in kilotons of C per square kilometer of the cell area).The sum of the potential carbon stocks for all cells north of 45ºN gives the conductivity-dependent estimate of the potential carbon stock in northern peatlands, which is equal to 875 PgC.

Since northern peatlands have already accumulated 547±74 PgC (Yu, 2011), the conductivity-dependent estimate of their potential carbon stock suggests that the total amount of carbon that they could remove from the atmosphere during the period from present to the end of the current interglacial is limited to 328±74 PgC.

The full range of uncertainty for the estimate of the amount of carbon that northern peatlands may accumulate from the start to the end of the current interglacial could be characterised by the uniform and clumped estimates. The former is equal to 665 PgC, and the latter is equal to 1258 PgC. However, our study shows that neither uniform interpretation nor clumped interpretation of the data on peatland extent is applicable everywhere, and hence the most likely range of uncertainty could be narrower than 665-1258 PgC.

## 4 Discussion

The limits to northern peatlands carbon stock were estimated here for the first time in the literature, although the methodology for obtaining such estimate were developed more than 30 years ago by Clymo (1984). We adapted this

methodology for use at the Earth system scale based on gridded data (Hengl et al., 2014) representing geomorphological aspects of peat bog growth.

We also characterized the uncertainty in the estimate of the limits to northern peatlands carbon stock induced by sub-grid distribution of peatland. This uncertainty cannot be easily reduced by using a finer grid, because it cannot be expected that each watershed falls within one grid cell. Therefore, we elaborated an approach for reducing uncertainty in the spatial distribution of peatlands that allows us to make a conclusion about the most likely value, 875 PgC, for this estimate.

Analyzing the uncertainty in the data on present-day peatland extent goes beyond the scope of this study. Improving the accuracy of these data is a well known task tackled by ISRIC, the International Soil Reference and Information Centre, (Batjes, 2016; Hengl et al., 2014), and by networks of peatland scientists such as C-Peat (Treat et al., 2019) and PeatDataHub (Xu et al., 2018). Hence, it might be more important to update the estimates of potential carbon stocks on a regular basis to keep pace with improvements in the accuracy of the data on present-day peatland extent.

The results of our study suggest that even the uniform estimate of the potential carbon stocks (665 PgC) is still higher than Gorham's (1991) estimate of 455 PgC in the actual carbon stocks of northern peatlands. Gorham's estimate, based on peat-volume approach (Loisel et al., 2014), is the product of the four numbers: mean depth of peatlands (2.3 m), mean bulk density of peat (112 Kg $m^{-3}$), carbon content of its dry mass (0.517), and the area of peatlands ($3.42 \times 10^{12}$ $m^2$). Our uniform estimate of potential carbon stocks implies that the potential mean depth of peat could be as high as 4 m for the same values of mean bulk density of peat and carbon content of its dry mass, and for smaller area of peatlands ($2.86 \times 10^{12}$ $m^2$). The uniform estimate is also higher than the Yu's (2011) estimate of actual carbon stocks, 547±74 PgC, based on the time history approach (Yu et al., 2010), suggesting that northern peatlands in total would accumulate in the future more carbon than they store now.

The clumped estimate, 1258 PgC, is beyond the range of uncertainty, 760-1006 PgC, in the estimate of potential carbon stocks that could be derived using the Yu's (2011) model of peat accumulation (see Supplement). Hence, it is reasonable to agree that the estimate of 875±125 PgC, as obtained from two completely independent methods, is the most expedient estimate of potential carbon stocks in northern peatlands, and that 330±200 PgC is the most expedient estimate of the total amount of carbon that they could remove from the atmosphere during the period from present to the end of the current interglacial.

The estimate of potential carbon stocks, 875±125 PgC, corresponds to the present climate, and therefore assumes that the present climate is typical for the present interglacial period. This assumption, however, might not be relevant to the scenarios of dramatic changes in the Earth system, jeopardizing peatlands development. The recent analysis of mitigation pathways compatible with global warming of 1.5°C above pre-industrial levels (Rogelj et al., 2018) shows that holding the global average temperature increase to well below 2°C is difficult but not impossible. To achieve this goal, cumulative $CO_2$ emissions from the start of 2018 until the time of net zero global emissions must be kept well below 1430 $GtCO_2$, (i.e., 390

PgC), that corresponds to 66[th] percentile of transient climate response to cumulative carbon emissions (Rogelj et al., 2018; Table 2.2). Since cumulative $CO_2$ emissions through to year 2017 are estimated at 610 PgC (Le Quéré et al., 2018), 1000 PgC of cumulative carbon emissions, the sum of historical (610 PgC) and the future cumulative emissions compatible with the global average temperature increase to below 2°C (390 PgC) could be considered as a threshold for defining the range of validity of the most expedient estimate of potential carbon stocks in northern peatlands. In brief, if cumulative carbon emissions do not exceed 1000 PgC, the northern peatlands play an important role in global carbon cycle recovery.

The ultimate recovery of the global carbon cycle from anthropogenic emissions is a long-term process (Archer, 2005). The current understanding of this process suggests that oceans absorb the majority of cumulative carbon dioxide emission within several centuries, the minor portion within several thousand years, and the remaining part will be removed through weathering of silicate rocks that may take hundreds of thousands of years (Archer, 2005; Archer and Brovkin, 2008; Brault et al., 2017). In other words, the larger the perturbation of the Earth system, the lower the chances that the pre-industrial state will be restored in course of the current interglacial.

Including peatlands in the consideration of global carbon cycle recovery allows us to evaluate the level of the Earth system perturbation that would not last too long to "break" the glacial-interglacial cycle. The results of numerical experiments (see Supplement) performed using an Earth system model of intermediate complexity (Brovkin et al., 2016) imply that keeping cumulative carbon dioxide emissions below 1000 PgC essentially reduces the risk of human intervention of natural glacial-interglacial cycle (Figure 4). The northern peatlands are capable to remove in relevant time frame, that is, over the next 5-15 thousand years, the amount of carbon that ocean will not able to remove, and thus to reduce the atmospheric carbon dioxide concentration to the level that is typical of interglacial periods.

**5 Conclusions**

Northern peatlands accumulate organic carbon and serve as a slow but persistent land carbon sink since the beginning of the current interglacial. If there were no limits to their growth in the absence of anthropogenic or natural $CO_2$ sources to the atmosphere, they could eventually reduce the atmospheric carbon dioxide concentration to the level at which a next precession-driven decline in the summer insolation in the high northern latitudes would trigger the onset of glaciation.

Our study, however, shows that the cumulative carbon removal associated with the natural development of peatland ecosystems is limited. The most expedient estimate of its potential magnitude, 875±125 PgC, was obtained under the assumption that the present climate is somewhat typical for the current interglacial period. Unless future scenarios of changes in the Earth system would leave no room for northern peatlands, the northern peatlands will play an important role in global carbon cycle recovery from anthropogenic emissions. While studies of this process are now focused on the strength and capacity of the ocean carbon sink, our results open a new perspective for the research on global carbon cycle recovery and on the measures needed to protect the northern peatlands as an important element of the Earth's climate system.

**Data availability.** All data used in this study are available from public databases or literature, cited in the Methods section. The data produced in course of this work are available from Georgii Alexandrov (g.alexandrov@ifaran.ru) upon request.

**Author Contributions.** All authors contributed to the conception of the work, to data processing and to writing of the paper. G.A.A. drafted the manuscript with inputs from V.A.B., T.K., and Z.Y.

**Competing interests.** The Authors declare no conflict of interests.

**Acknowledgements.** G.A.A. acknowledges funding by RFBR according to the research project № 19-05-00534. The manuscript has been initiated during a visit of G.A.A. to the Land in the Earth System Department of the Max Planck Institute of Meteorology in 2017. The comments of anonymous reviewers helped us to improve the original manuscript.

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

**Table 1. Potential peat carbon density at the central part of peatland estimated under uniform (PCD1) and clumped (PCD2) interpretation of $f_{P, obs}$ as compared to the observed peat carbon density (PCD0) at 33 peatland sites (Yu et al., 2009).**

| Site # | Region | Type | Location | PCD0 kgC m$^{-2}$ | PCD1 kgC m$^{-2}$ | PCD2 kgC m$^{-2}$ | $f_{P, obs}$ % |
|---|---|---|---|---|---|---|---|
| 1 | West Siberia | bog | 60º10'N 72º50'E | 230 | 1148 | 2239 | 56 |
| 2 | West Siberia | bog | 60º10'N 72º50'E | 268 | 1148 | 2239 | 56 |
| 3 | West Siberia | bog | 56º50'N 78º25'E | 413 | 1277 | 1432 | 72 |
| 4 | West Siberia | fen | 56º20'N 84º35'E | 399 | 849 | 1444 | 60 |
| 5 | Alaska | fen | 60º27'N 151º14'W | 149 | 190 | 1437 | 20 |
| 6 | Alaska | fen | 60º38'N 151º04'W | 142 | 191 | 1449 | 20 |
| 7 | Alaska | rich fen | 60º25'N 150º54'W | 117 | 157 | 1155 | 20 |
| 8 | Alaska | poor fen | 60º47'N 150º49'W | 64 | 219 | 1687 | 20 |
| 9 | Alaska | taiga bog | 64º52'N 147º46'W | 133 | 102 | 692 | 20 |
| 10 | Canada | slope bog | 54º09'N 130º15'W | 73 | N/A | N/A | 0 |
| 11 | Canada | rich fen | 53º35'N 118º01'W | 232 | 68 | 864 | 10 |
| 12 | Canada | fen | 52º27'N 116º12'W | 317 | 55 | 623 | 10 |
| 13 | Canada | bog | 55º01'N 114º09'W | 228 | 1499 | 1811 | 70 |
| 14 | Canada | permafrost | 61º48'N 121º24'W | 147 | 72 | 566 | 16 |
| 15 | Canada | fen | 68º17'N 133º15'W | 61 | 82 | 524 | 20 |
| 16 | Canada | fen | 69º29'N 132º40'W | 27 | N/A | N/A | 0 |
| 17 | Canada | permafrost | 55º51'N 107º41'W | 141 | 99 | 1294 | 11 |
| 18 | Canada | fen | 64º43'N 105º34'W | 65 | N/A | N/A | 0 |
| 19 | Canada | fen | 66º27'N 104º50'W | 84 | N/A | N/A | 0 |
| 20 | Canada | permafrost | 59º53'N 104º12'W | 81 | N/A | N/A | 0 |
| 21 | Canada | bog | 45º41'N 74º02'W | 70 | N/A | N/A | 0 |
| 22 | Canada | rich fen | 82°N 68°W | 97 | N/A | N/A | 0 |
| 23 | Canada | N/A | 47º56'N 64°30'W | 275 | 58 | 678 | 10 |
| 24 | Canada | N/A | 45º56'N 60º16'W | 209 | 54 | 606 | 10 |
| 25 | Scotland | bog | 57º31'N 5º09'W | 106 | N/A | N/A | 0 |
| 26 | Scotland | bog | 57º34'N 5º22'W | 195 | 129 | 873 | 21 |
| 27 | Scotland | bog | 57º41'N 5º41'W | 151 | 160 | 493 | 40 |
| 28 | Finland | palsa mire | 68º24'N 23º33'E | 122 | 190 | 1438 | 20 |
| 29 | Finland | fen | 68º24'N 23º33'E | 134 | 190 | 1438 | 20 |
| 30 | Finland | raised bog | 60º49'N 26º57'E | 214 | 65 | 1350 | 6 |
| 31 | Finland | aapa mire | 65º39'N 27º19'E | 123 | 499 | 994 | 55 |
| 32 | Finland | aapa mire | 65º39'N 27º19'E | 154 | 499 | 994 | 55 |
| 33 | Finland | fen | 65°39'N 27º19'E | 215 | 499 | 994 | 55 |

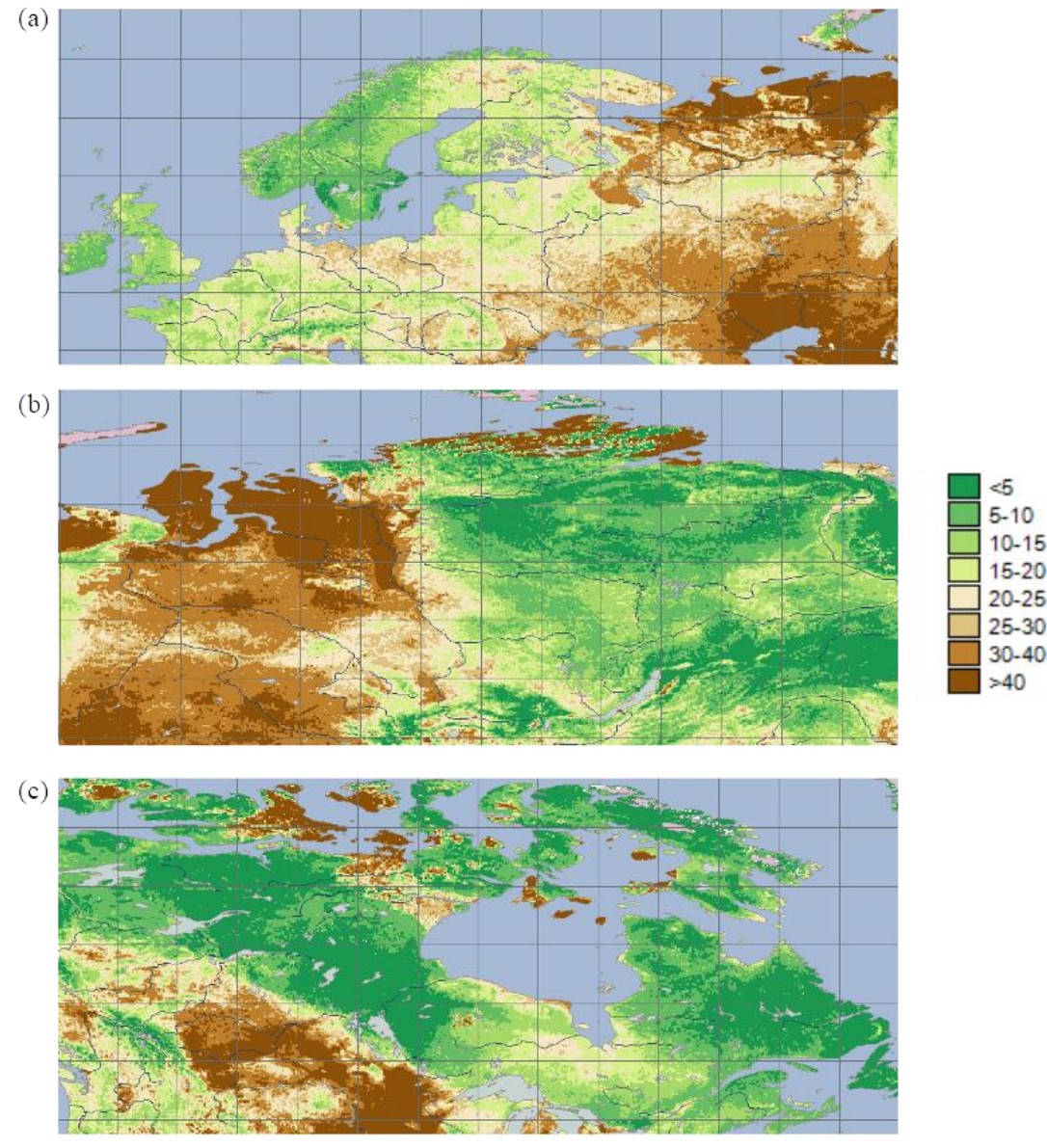

**Figure 1: The depth to bedrock, an estimate of *g*, in meters, in Europe (a), Western Siberia (b), Canada (c).**

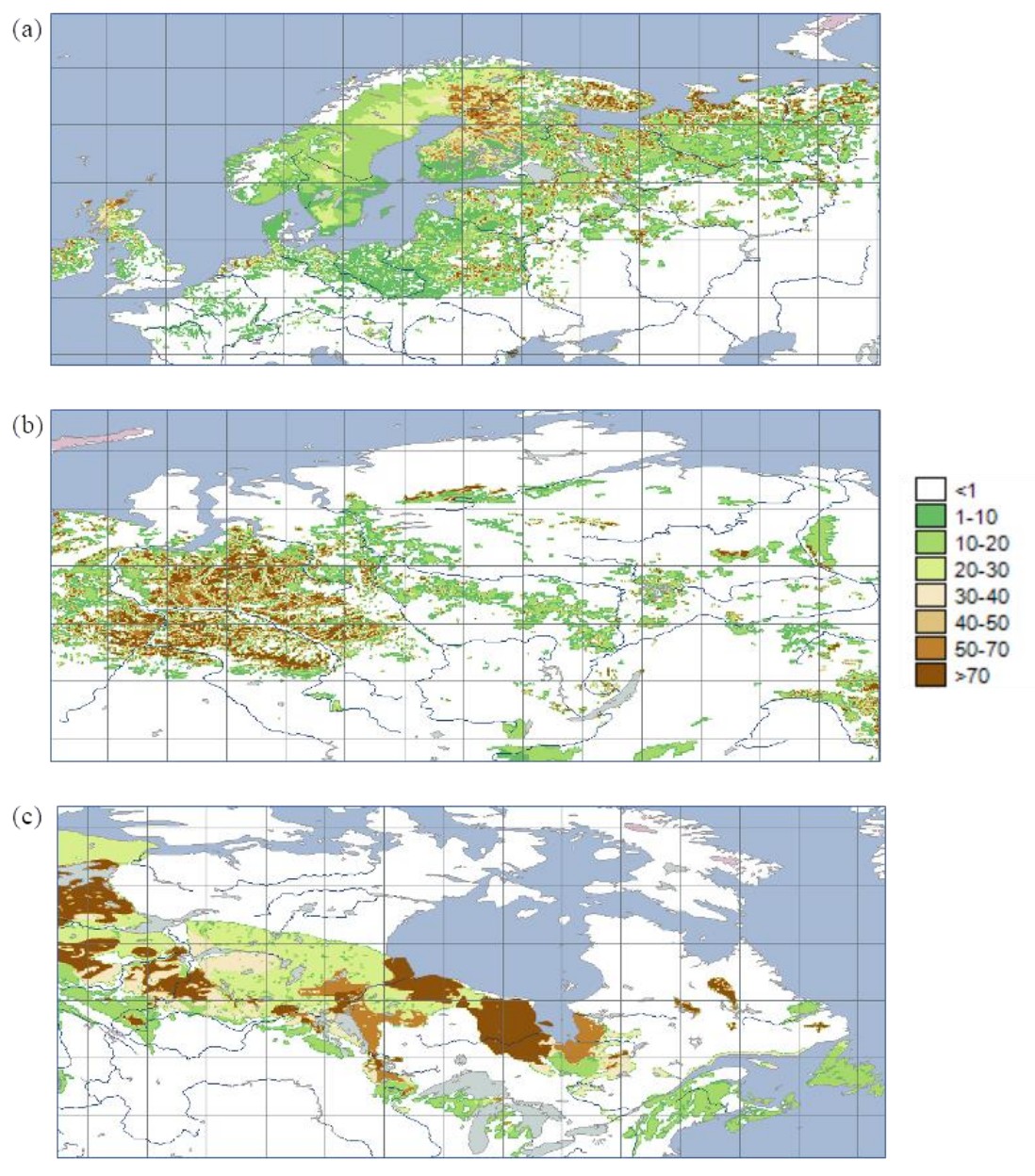

**Figure 2: The fraction of histosols (%) in Europe (a), Western Siberia (b), and Canada (c).**

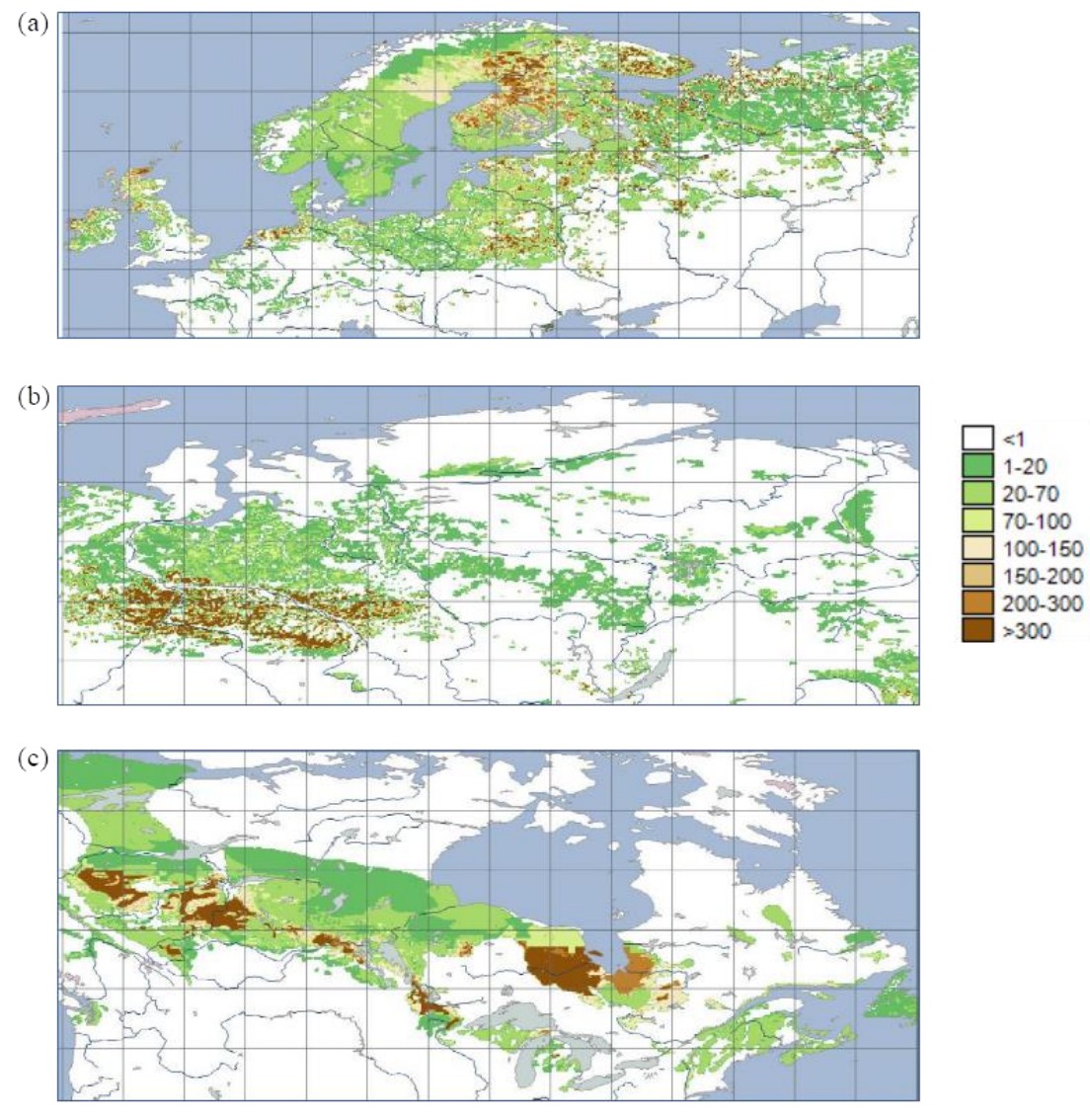

**Figure 3: The conductivity-dependent estimate of the potential carbon stocks in northern peatlands per area of a grid cell (x10$^9$ gC km$^{-2}$) in Europe (a), Western Siberia (b), and Canada (c).**

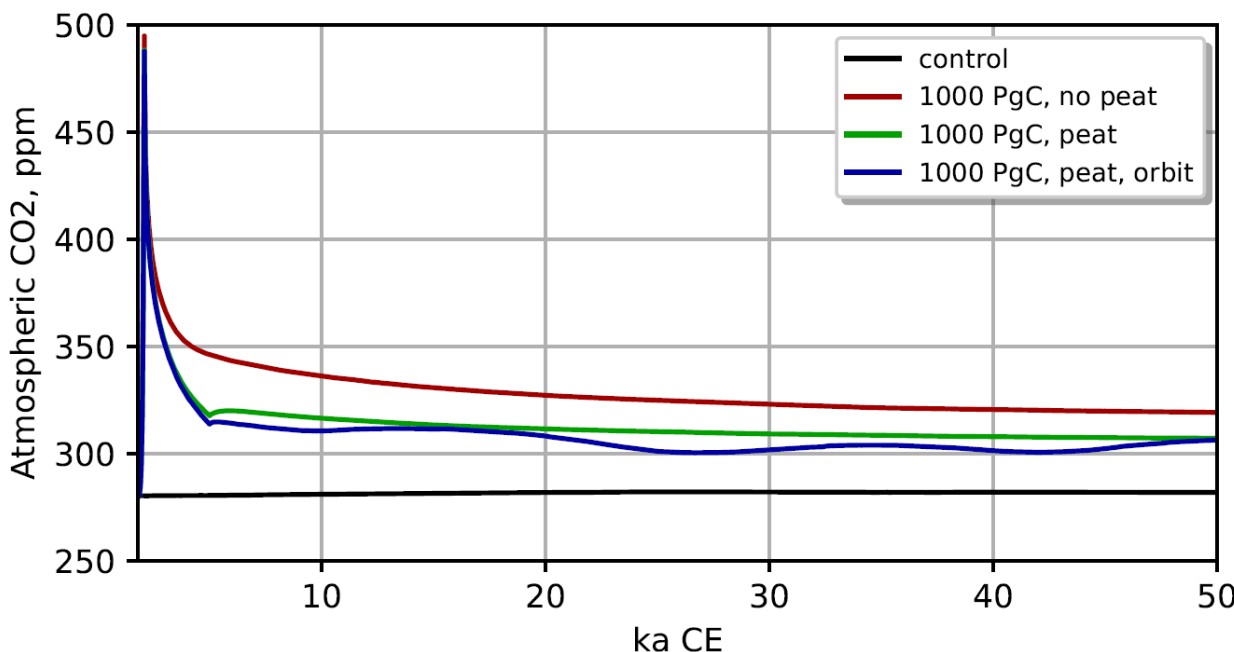

5  **Figure 4: Multimillennial changes in the atmospheric CO₂ concentration simulated using CLIMBER-2, an Earth system model of intermediate complexity (Brovkin et al., 2016), for scenario of 1000 PgC cumulative emissions. No peatlands (mainly ocean CO₂ uptake, red line), plus northern peatlands uptake of 330 PgC (green line), plus orbital forcing effect (blue line).**

