# Peer review of "The limits to growth of northern peatland carbon stocks"

_Biogeosciences, 2019_

## Referee Comment (RC1) · Anonymous Referee #1 · 23 May 2019

Alexandrov et al. raise an interesting topic and modeled the potential for carbon sequestration in northern peatlands. They show that large amounts of carbon in the atmosphere could be offset by peatland growth throughout the current interglacial. I think the study focuses on an important topic and the results are worth publishing, however, the methods and the results need to be presented in a revised, more precise and coherent form. I think the paper should be significantly revised before consideration of publication. Please see my detailed comments below.

Abstract:

Please insert one or two statements about the methods, which you applied in this study. Also, include a statement about your results, where you specifically mention the amount of carbon which could be set off by peatland growth.

In addition, I would recommend changing the title of the manuscript into "The potential of northern peatlands for carbon sequestration"

Specific comments:

Page 1, Line 10: Maybe write "continuous" instead of "persistent"

P1, L.12: Rewrite the sentence. E.g. "The evaluation of the carbon sequestration potential of northern peatlands show that atmospheric carbon dioxide concentration can be significantly reduced. Northern peatlands have the potential to be the second largest $CO_2$ sink after the world's oceans."

Introduction:

General comments:

The introduction needs a better structure. The different paragraphs need to be connected better and the research gap should be mentioned more clearly. Also, state in the end of the introduction what the goals of your study are. The last two paragraphs (p2, line 17-29) belong into the methods part and should be removed from the introduction.

Specific comments:

Page 1, Line 17: You mention the study by Loisel et al. (2014). Please also include the new study by Treat et al. (2019) in your introduction

P1, L.17 : I suggest to use the word "knowledge" instead of "wisdom"

P1, L.21 : I suggest to use the word "previous" instead of "later"

P1, L.23 : Where do northern peatlands start? Is it >40° North or >45° North, please clarify

P1, L.25: 864-2240 PgC – Is that already your result or is it from a different study – please clarify

P2, L.13: I suggest to use the word "rise" instead of "elevation"

Methods:

General comments:

The methods are somewhat unclear to me. You start with an explanation of the maximum depth of peat, however in equation 1 you show how the maximum C stock can be calculated. You could start with an equation for the maximum depth of peat before introducing the maximum carbon stock in a grid cell. In addition, I suggest to make subchapters to explain the different model parameters. The first subchapter could include the maximum carbon stock in a grid cell, whereas a second subchapter includes the extrapolation from the grid cell to the entire northern peatland area and a third subchapter explains the differences between a conservative and non-conservative interpretation of fp. Also, in the end of the methods, it appears to be a mix of discussing your methods and presenting some results already. I suggest you discuss your methods in the discussion section with a separate subchapter and strictly separate between methods and results, so that no results appear in the methods section.

Specific comments:

P3, L.3: What is the density of draining system – please explain

P3, L.4: What is the impeded drainage model? – If this is your own model, you should explain it in the methods, otherwise add a reference.

P3, L.6: I do not understand the second, smaller equation. Why is hmax, the maximum height of the water table above the level of the draining system, dependent from the fraction of the area occupied by peatlands?

P3, L.9: Change the sentence to "...hmax is the maximum height of the water table above the level of the draining system..."

P3, L.26: How much is the minimal depth of the peat layer which is used to classify a

land unit as peatland? – Please give a number or a range for the minimal peat depth.

P4, L.1-11: This part would better fit into the discussion where you could have a sub-chapter discussing your methods and you model.

P4, L.12: What is the non-conservative and what is the conservative interpretation of $f_{p,obs}$, please add values

P4, L.13: 1258 vs 665 PgC. This is a result and should therefore be in the results chapter

P4, L.14: Please replace "one cannot expect. . ." with "it cannot be expected. . ."

P4, L.18: Please replace "one may assume. . ." with "it can be assumed. . ."

P4, L.21: "peat C addition" do you mean C accumulation?

P4, L.23: 875 PgC. This is another result and should therefore be in the result section.

Results

General comments:

Please present here your own results and do not start with a comparison to another study. Instead of all the numbers from Gorham (1991), present your own results for mean depth of peatlands, mean bulk density or area of peatlands. The comparison with Gorham (1991) as well as Yu (2011) belongs to the discussion part. The results section needs to be rewritten completely with a focus on your own results.

Specific comments:

P5, L.5: Add the year of publication after Yu

P5, L.10: Add the year of publication after Yu

P5, L.10: Please change "one could find" into "it is reasonable to agree. . ."

P5, L.11: Why 875 PgC? What is with the 665 PgC – 1258 PgC? What is your main

result? This needs to be clear.

Discussion:

General comments:

The discussion is very short. Please provide a more in-depth discussions of your methodological approach, e.g. show the benefits but also limitations of your model and compare your results of potential C accumulation with e.g. C accumulation during the Holocene. I suggest making several subchapters. One where you discuss the benefits and limitations of your model, including the uncertainty of your estimation. Another subchapter where you compare your results with previous studies (as you did in the results section) and a third subchapter where you discuss the implications of your results on the global C cycle (basically your actual discussion).

Specific comments:

P5, L.14: Change the first sentence into: "The potential for northern peatlands to store carbon were estimated for the first time..."

P5, L.15f: Change the following sentence into: "We adapted this methodology to global scale and additionally included geomorphological aspects of peat bog growth..."

P5, L.18 : Write "Our estimate..." instead of "Moreover, this..."

P5, L.18: Delete "somewhat"

P5, L.19: Change the following sentence into: "This assumption, however, might not be relevant for scenarios of dramatic changes in the Earth system that will take place if cumulative carbon..."

P5, L.20-21: Why 1000 PgC? It seems a bit arbitrary to me? Can you discuss this a bit more?

P5, L.21f: Change the following sentence into: "Nevertheless, if cumulative carbon

emissions do not exceed 1000 PgC, the northern peatlands play an important role in global carbon cycle recovery"

P5, L.21: What happens to the peat C storage if carbon emissions exceed 1000 PgC?

P5, L.26 : Replace "plain" with "other"

P6, L.1-4: You should also discuss the conditions and timeframe under which such a scenario can happen. Is this only under ideal conditions? What about the limitations in the model? Also, if you make such a strong statement, there should be a better explanation of this Earth system model of intermediate complexity.

P6, L.2: Maybe you can elaborate a bit more on figure 4. How does the orbital forcing affect peatland C uptake?

P6, L.2: "in relevant time frame" – Can you give a number, what a relevant time frame is?

P6, L.3 replace "won't" with "will not be able to"

Conclusions:

P6, L.7: What are limits to peatland growth? – Please discuss this in the discussion section

P6, L.10-16: This section is somewhat contradictorily in itself and compared to other parts of the manuscript. Why is the cumulative carbon removal associated with the natural development of peatland ecosystems limited? – Please discuss this in the discussion section

Supplement

Please add a reference list for the supplement

S1.1 : Please rephrase the first sentence.

References:

Gorham, E.: Northern peatlands: role in the carbon cycle and probable responses to climatic warming, Ecol. Appl., 1(2), 182–195, doi:10.2307/1941811, 1991

Loisel, J., Yu, Z., Beilman, D. W., Camill, P., Alm, J., Amesbury, M. J., Anderson, D., Andersson, S., Bochicchio, C., Barber, K., Belyea, L. R., Bunbury, J., Chambers, F. M., Charman, D. J., De Vleeschouwer, F., Fiałkiewicz-Kozieł, B., Finkelstein, S. A., Gałka, M., Garneau, M., Hammarlund, D., Hinchcliffe, W., Holmquist, J., Hughes, P., Jones, M. C., Klein, E. S., Kokfelt, U., Korhola, A., Kuhry, P., Lamarre, A., Lamentowicz, M., Large, D., Lavoie, M., MacDonald, G., Magnan, G., Mäkilä, M., Mallon, G., Mathijssen, P., Mauquoy, D., McCarroll, J., Moore, T. R., Nichols, J., O'Reilly, B., Oksanen, P., Packalen, M., Peteet, D., Richard, P. J. H., Robinson, S., Ronkainen, T., Rundgren, M., Sannel, A. B. K., Tarnocai, C., Thom, T., Tuittila, E. S., Turetsky, M., Väliranta, M., van der Linden, M., van Geel, B., van Bellen, S., Vitt,D., Zhao, Y. and Zhou, W.: A database and synthesis of northern peatland soil properties and Holocene carbon and nitrogen accumulation, Holocene, doi:10.1177/0959683614538073, 2014.

Treat, C. C., T. Kleinen, N. Broothaerts, A. S. Dalton, R. Dommain, T. A. Douglas, J. Z. Drexler, S. A. Finkelstein, G. Grosse, G. Hope, J. Hutchings, M. C. Jones, P. Kuhry, T. Lacourse, O. Lähteenoja, J. Loisel, B. Notebaert, R. J. Payne, D. M. Peteet, A. B. K. Sannel, J. M. Stelling, J. Strauss, G. T. Swindles, J. Talbot, C. Tarnocai, G. Verstraeten, C. J. Williams, Z. Xia, Z. Yu, M. Väliranta, M. Hättestrand, H. Alexanderson & V. Brovkin (2019) Widespread global peatland establishment and persistence over the last 130,000 y. Proceedings of the National Academy of Sciences, 116, 4822, doi: 10.1073/pnas.1813305116, 2019

Yu, Z.: Holocene carbon flux histories of the world's peatlands: Global carbon-cycle implications, The Holocene, 21(5), 761–774, doi:10.1177/0959683610386982, 2011.

---

## Referee Comment (RC2) · Anonymous Referee #2 · 24 May 2019

Journal: Biogeosciences

Manuscript no. : bg-2019-76

Author(s): Alexandrov et al.

Date submitted: 20th March 2019

General comments: In this manuscript, Alexandrov et al. present and discuss the estimates of northern peatlands carbon stocks using different approaches (conservative, non- and less-conservative approach). The procedure to calculate the total carbon content for the northern peatland areas have already been developed but in this study, authors have revised some values which they have estimated using the gridded soil dataset. The study has the potential to reduce the current uncertainties related to the

limits of peatland carbon stocks and it is worth publishing. However, I find there are many sections which need to be strengthened, particularly, the methodology and result sections. I also recommend them to divide the methods section into several parts under different sub-headings and include a brief explanation about the model in the beginning. In the introduction and discussion sections, many arguments need to be referenced (see my comments below). More importantly, the authors have assumed that peatland distribution areas have not much been changed since the last 5000 years and the growth in the peat height was a major cause of carbon uptake in the northern areas. However, according to MacDonald et al. 2006 (see Figs. 1 and 3), around 30-40% peatlands were initiated after 5000 cal. B.P. which means that the increase in new peatland areas has also played a significant role in sequestering atmospheric carbon. How do they explain this assumption?

Specific comments:

P1 L18: How did you define northern peatlands (> 40°N or 45° N)?

P1 L19: "The variations are explained by" . . . Which variations?

P1 L22: "However, during the last 5000 years, the area of peatlands remained relatively stable. . ."

Peat basal ages are used as proxies to identify new peatland areas and expansion rate. From figures 1 and 3 in MacDonald et al. 2006, we can see that around 30-40% of the peatlands were initiated after 5000-year cal. B.P in northern areas. Therefore, I doubt whether the growth in the peat depth is the only major cause of carbon uptake in the past.

P1 L25: "the northern peatlands may accumulate 864-2200 PgC . . ."

This is a very high value, how did you calculate this range. From where did you find this information? What about the peatland distribution area and sink capacity, will they remain the same in the future? Studies indicated that many peatlands would lose their

carbon sink capacity while some may enhance.

P2 L 1-15: Support your arguments with previously established knowledge. Include references.

P2 L5: Define what a steady state is for your readers.

P2 L6: How did you estimate this range – see my previous comment.

P2 L9: Remove this expression – "the so called"

P2 L11: "at least a small portion of the organic matter that enters the acrotelm always reaches to the catotelm . . ."

Is this a plausible argument – do you think, acrotelm always passes organic matter in the catotelm? Even when peatland experiences continuous dry conditions?

P2 L 13-15: In which study, did you find this information?

P2 L 16: Could you explain a bit about your model. What it does and other relevant information briefly and give more details in the methods section.

P2 L 22: "The gridded data on soil properties give the fraction of a grid cell covered by peatlands . . ."

Include reference.

P2 L27: How did you determine where to form a cluster in a grid cell?

P2: I think a paragraph needs to be added in the end which explains the purpose of your study.

P3 L1: Methods

Perhaps subheadings could be helpful to improve and clarify the structure of the methods. I also suggest you to add a model description section.

P3 L3: "The density of the draining system" – explain what it is?

P3 L4: "The impeded drainage model approach" – Give more details about this approach and model. What it is and where this approach has been used before?

P3 eqn 1 - From where this equation comes from? Any previous applications? Please clarify.

P3 L7-8: it is better to include the value of constants in the equation or under it.

P3 L10: There are many peatlands in the southern latitude region between 45-55°N, particularly in China, U.S and Magnolia. Have you considered them in your calculation?

P3 L11: Include a brief write up about the SoilGrid dataset and what it contains.

P3 L18: Did you check the recent study by Xu et al. 2018 where the authors have refined the global and regional estimates of peatland distribution area? How your dataset (WISE30sec) is different or better than Xu et al. 2018 (PEATMAP)?

P3 L22: If you have the dataset then you can easily estimate how much area is occupied by northern peatlands. According to Xu et al., around 3.12 million km2 area is occupied by peatlands above 45°N and Yu et al. 2010, used 4.0 million km2.

P4 L: How accurate are these conservative and non-conservative estimates? From Table 1, one can see that both estimates fail to capture the observed peatland carbon density. In fact, in some cases, the conservative estimates are higher than the observed values. Based on this information, do you think we can rely on your modelled limits?

P4 Results: This looks like a part of the discussion. I recommend you to explain your results and what you see in your figures before comparing them with the previously established knowledge. For example, you can highlight how much peatland carbon stocks you have estimated in the European, Russian and N. American regions, which areas are rich in carbon within in these regions, what are the maximum and minimum peat depths, why you used a constant bulk density value etc.

P4 L27: You can also include the eqn used by Gorham 1991- Cpeat = Pi (Ai × Di × BDi × CCi)

P4 L29: 112 × 103 g m-3 - Change it to 112 kg m-3

P5 L4: Explain what time history approach is.

P5 L15: There are other methodologies which have been developed to estimate total carbon stocks (see Yu et al. 2012). How your approach is different or better than these methodologies and what are its limitations?

P5 L18-22: No references.

P5 L16: "We adapted this methodology for use at the global scale . . ."

Global or regional because you have considered only the northern peatlands?

P6 L7: "If there were no limits to their growth. . ."

In the introduction, you have mentioned that peatlands can reach to steady state and do not grow or accumulate carbon after that. Do your analysis shows in which regions peatlands have already reached to steady state?

References: 1. MacDonald GM, Beilman DW, Kremenetski KV, Sheng Y, Smith LC, Velichko AA. Rapid Early Development of Circumarctic Peatlands and Atmospheric CH4 and CO2 Variations. Science 314, 285-288 (2006).

2. Gorham E. Northern Peatlands - Role in the Carbon-Cycle and Probable Responses to Climatic Warming. Ecol Appl 1, 182-195 (1991).

3. Xu JR, Morris PJ, Liu JG, Holden J. PEATMAP: Refining estimates of global peatland distribution based on a meta-analysis. Catena 160, 134-140 (2018).

4. Yu ZC. Northern peatland carbon stocks and dynamics: a review. Biogeosciences 9, 4071-4085 (2012).

---

## Author Comment (AC1) · 18 Jun 2019

We are pleased to see that Referee agrees that the results of the study are worth publishing. We also acknowledge the importance of Referee's comments for presenting the results of the study in a more precise and coherent way. Below is our response to Referee's questions and recommendations.

Response to the questions asked by Referee:

1. (1) P1, L.23 : Where do northern peatlands start? Is it >40 North or >45

(2) Here we refer to the article of Loisel et al. (2017) and keep in mind the peatlands located north of 45 N.

[Figure]

(3)ãĂĂ". . . northern peatlands, namely the peatlands distributed across the northern mid- and high-latitude regions located north of 45°N, . . ."

2. (1) P1, L.25: 864-2240 PgC – Is that already your result or is it from a different study?

(2) These numbers were calculated from the range of estimates of carbon accumulation rates associated with peat growth and the range of estimates of peatland area reported by Yu (2011) and cited in this paragraph. It is a starting point of our research aimed to explore limitations to peatlands growth that do not allow them to remove 2000 PgC amount of carbon from the atmosphere.

(3) The changes made in the manuscript hopefully make it clear that this is a simple extrapolation based on the estimates reported by Yu (2011).

3. (1) P3, L.3: What is the density of draining system?

(2) The density of draining system is the length of draining streams per unit area.

(3) ". . .the potential peat depth, is determined by the amount of effective rainfall, drainage system density (the length of draining streams per unit area) and the hydraulic conductivity . . ."

4. (1) P3, L.4: What is the impeded drainage model?

(2) The impeded drainage model is the model based on the Dupuit-Forchheimer theory of groundwater movement (aka hydraulic theory) and a few additional assumptions (see Supplementary Information to our previous work, https://media.nature.com/original/nature-assets/srep/2016/160420/srep24784/extref/srep24784-s1.doc). The basic idea of the model is that the high level of water table in a peatland is maintained due to impeded drainage: it takes long time for water coming with precipitation at the central part of a peatland to reach the draining streams.

(3) "To calculate the potential peat depth, we apply an equation derived (see Supplement) from the impeded drainage model used in our previous study (Alexandrov et al., 2016)?."

5. (1) P3, L.6: Why is hmax, the maximum height of the water table above the level of the draining system, dependent from the fraction of the area occupied by peatlands?

(2) Both hmax and fp, the fraction of the area occupied by peatlands, depend on K, the hydraulic conductivity: equation (S6) and equation (S10) in the Supplement (https://www.biogeosciences-discuss.net/bg-2019-76/bg-2019-76-supplement.pdf). The "observed" value the fraction of the area occupied by peatlands, f_P,obs makes it possible to estimate K: equation (S11). Substituting K given by equation (S11) to equation (S6) gives the equation (S12), where hmax depends on f_P,obs and g, the average height of the watershed above the level of the draining system. That is to say, excluding K from the equation for hmax leads to including fP,obs into this equation.

(3) This part of the text is rewritten.

6. (1) P3, L.26: How much is the minimal depth of the peat layer which is used to classify a land unit as peatland?

(2) The minimal depth of the peat layer which is used to classify a land unit as peatland is a source of uncertainty in the estimates of peatland area. We relied on the WISE30sec data set (Batjes, 2016) of soil properties and diagnosed peatland extent by fraction of grid cell covered by soils of histosol type. Hence, the minimal depth of the peat layer is assumed to be 40 cm (according to FAO definition of histosols).

7. (1) P4, L.12: What is the non-conservative and what is the conservative interpretation of fp,obs?

(2) The variety of possible interpretations is parameterized using the equation (S18) in the Supplement. The difference between the conservative and non-conservative interpretations of fp,obs could be illustrated by the following example. Let us consider a grid cell the 36% of which is covered by peatlands. Does it mean that peatlands cover 36% of each watershed within this grid cell? Or does it mean that only 48% of watersheds are occupied by peatlands, and the peatlands cover 75% (0.48*75=36) of each of these watersheds? In other words, we cannot say for sure whether the grid cell contains many small peatlands, or few large peatlands. Under the conservative inter-pretation, fp,obs = 36% suggests that peatlands cover 36% of each watershed within this grid cell (many small peatlands). Under the non-conservative interpretation, fp,obs = 36% suggests that only 48% of watersheds within the grid cell are occupied by peat-lands, and the peatlands cover 75% (0.48*75=36) of each of these watersheds (few large peatlands). The conservative interpretation of fp,obs leads to smaller estimate of pmax as compared to the non-conservative interpretation.

(3) This part of the text is rewritten.

8. (1) P5, L.11: Why 875 PgC? What is with the 665 PgC – 1258 PgC? What is your main result?

(2) The main result of this study is the expedient estimate of carbon stock that could be accumulated by northern peatlands by the end of the current interglacial. This estimate is equal to 875 PgC and falls within the range of uncertainty that starts from 750 PgC to 900 PgC (= 875±125 PgC), and derived from the Yu's model (Yu, 2011). The validity of this estimate is supported by the estimates of potential carbon stocks obtained by a completely independent method under different interpretations of the data on the georgraphic distribution of peatlands: this estimate, 875±125 PgC, falls within the range of uncertainty associated with accuracy of the data on the georgraphic distribution of peatlands.

(3) This part of text is rewritten.

9. (1) P4, L.21: "peat C addition" do you mean C accumulation?

(2) We use the words "peat C addition" to denote the amount of carbon that enter to catotelm. Peat accumulation is the difference between peat addition and peat decomposition.

(3) The words "peat C addition" are changed to "annual C input to catotelm." "This model suggests that the growth of carbon stock in peatlands is limited by the ratio of annual C input to catotelm to the decay constant."

10. (1) P5, L.20-21: Why 1000 PgC? It seems a bit arbitrary to me.

(2) It is not completely arbitrary. According to Allen et al. (Nature. 2009 Apr 30;458(7242):1163-6. doi: 10.1038/nature08019), cumulative anthropogenic emissions of 1000 PgC are expected to result in 2oC carbon-dioxide-induced warming above pre-industrial temperatures (confidence interval: 1.3–3.9 oC). Hence, if cumulative anthropogenic emissions will not exceed 1000 PgC, then there is a chance that there will be no dramatic changes in climate leading to a massive destruction of northern peatlands.

(3) "This assumption, perhaps, is not relevant to the scenarios of dramatic changes in the Earth system, jeopardizing peatlands development, that might take place if cumulative carbon emissions exceed 1000 PgC (Allen et al., 2009; Millar et al., 2017)."

11. (1) P5, L.21: What happens to the peat C storage if carbon emissions exceed 1000 PgC?

(2) According to Millar et al. (Nature Geoscience, 10: 741-747.), 90% of CMIP5 models suggest that 468 PgC of cumulative carbon emissions after 2015 lead to warming by 1oC above 2010-2019 level under RCP2.6 scenario of radiative forcing. Hence, if cumulative carbon emissions exceed 1000 PgC (545 PgC from 1850 to 2015 plus 468 PgC after 2015), then one cannot exclude the risk of dramatic changes in climate leading to a massive destruction of northern peatlands.

(3) The text is revised.

12. (1) How does the orbital forcing affect peatland C uptake?

(2) We did not consider the effect of orbital forcing on peatland C uptake in the reported numerical experiment. The purpose of this experiment was to show how additional long-term carbon sink provided by northern peatlands may affect the level of atmospheric $CO_2$ concentration to which Earth system will return after the end of anthropogenic $CO_2$ emissions.

13. (1) P6, L.2: "in relevant time frame" – Can you give a number, what a relevant time frame is?

(2) The next reductions in northern summer insolation that may lead to glacial inception will occur 1500, 16000, and 53000 years after present. It is unlikely that atmospheric carbon dioxide concentration will return to the level typical for interglacial periods within next 1500 years. Hence, the next 5-15 thousand years is a relevant time frame for reducing the atmospheric carbon dioxide concentration to the level that is typical of interglacial periods.

(3) "in relevant time frame, that is, in 5-15 thousand years,"

14. (1) P6, L.7: What are limits to peatland growth?

(2) The limits to peatland growth are the peat depth values that cannot be exceeded in given climatic and geomorphologic conditions.

15. (1) P6, L.10-16: Why is the cumulative carbon removal associated with the natural development of peatland ecosystems limited? The cumulative carbon removal associated with the natural development of a peatland ecosystem is limited by the height of the water table that could be maintained due to impeded drainage above the level of draining streams. Therefore, cumulative carbon removal associated with the natural development of peatland ecosystems is limited by the given climatic and geomorphological conditions.

Response to recommendations made by Referees:

1. (1) Abstract: Please insert one or two statements about the methods, which you applied in this study. Also, include a statement about your results, where you specifically mention the amount of carbon which could be set off by peatland growth.

(2) done

(3) "The limits to northern peatland carbon stocks, evaluated based on the gridded data on the depth to bedrock and on the fraction of area covered by soils of histosol type, suggest that 875±125 PgC is the most expedient estimate of potential carbon stock in northern peatlands at large."

2. (1) I would recommend changing the title of the manuscript into "The potential of northern peatlands for carbon sequestration"

(2) We would like to keep a "connotation" to the paper "The limits to peat bog growth" by Clymo (1984), and to highlight the fact that the cumulative amount of carbon that northern peatlands could remove from the atmosphere is limited by the geomorphological conditions in present climate.

(3) Based on the overall idea of the suggested title, we think it would be reasonable to change the title as follows, "The limits to growth of northern peatland carbon stocks".

3. (1) Page 1, Line 10: Maybe write "continuous" instead of "persistent"

(2) We used "persistent" instead of "continuous", because "persistent carbon sink" is a common collocation appeared in a number of research articles on carbon cycle (e.g., Pan, Y. D. et al. A large and persistent carbon sink in the world's forests. Science 333, 988–993 (2011)).

4. (1) P1, L.12: Rewrite the sentence. E.g. "The evaluation of the carbon sequestration potential of northern peatlands show that atmospheric carbon dioxide concentration can be significantly reduced. Northern peatlands have the potential to be the second largest CO2 sink after the world's oceans."

(2) Here we were trying to say that over the next 5 thousand years after the end of fossil fuel burning, not only oceans but also northern peatlands will be removing carbon dioxide from the atmosphere.

(3) This part of the text is rewritten.

5. (1) The introduction needs a better structure. The different paragraphs need to be connected better and the research gap should be mentioned more clearly.

(2) Done

(3) We moved the paragraph that seemingly was breaking the logic flow to the proper place in the end of Introduction.

6. (1) Page 1, Line 17: You mention the study by Loisel et al. (2014). Please also include the new study by Treat et al. (2019) in your introduction

(2) Done

7. (1) P1, L.17 : I suggest to use the word "knowledge" instead of "wisdom"

(2) We think that "notion" would be good.

(3) "... largely confirm the conventional notion of the carbon (C) sink provided by northern peatlands ..."

8. (1) P1, L.21 : I suggest to use the word "previous" instead of "later"

(2) Here we use 'later' in sense of 'coming after something else'. Hence, it cannot be replaced by "previous".

(3) To avoid possible misunderstanding, we revise this sentence as follows, "In the early Holocene, both the rate of peatland expansion and the rate of carbon accumulation appear to be highest (Yu et al., 2010) as compared to the later Holocene periods."

9. (1) P2, L.13: I suggest to use the word "rise" instead of "elevation"

(2) Done

10. (1) Methods: You could start with an equation for the maximum depth of peat before introducing the maximum carbon stock in a grid cell

(2) Done

11. (1) I suggest to make subchapters to explain the different model parameters. The first subchapter could include the maximum carbon stock in a grid cell, whereas a second subchapter includes the extrapolation from the grid cell to the entire northern peatland area and a third subchapter explains the differences between a conservative and non-conservative interpretation of fp

(2) done

(3) The estimate for the entire peatland area is merely a sum carbon stocks in the grid cells located north of 45N, and hence is explained in one sentence, "The sum of the potential carbon stocks for all cells north of 45?N gives the less-conservative estimate of the potential carbon stock in northern peatlands."

12. (1) I suggest you discuss your methods in the discussion section with a separate subchapter and strictly separate between methods and results, so that no results appear in the methods section

(2) Done

(3) No results appear in the methods section. The discussion section is rewritten.

13. (1) P3, L.9: Change the sentence to ": : :hmax is the maximum height of the water table above the level of the draining system: : :" (2) done

14. (1) P4, L.1-11: This part would better fit into the discussion where you could have a subchapter discussing your methods and you model.

(2) This part is an explanation of our approach to addressing uncertainty. We rewrite

the text to clarify this point.

(3) This part of the text is rewritten.

15. (1) P4, L.13: 1258 vs 665 PgC. This is a result and should therefore be in the results chapter (2) done

16. (1) P4, L.14: Please replace "one cannot expect: : :" with "it cannot be expected: : :" (2) done

17. (1) P4, L.18: Please replace "one may assume: : :" with "it can be assumed: : :" (2) done

18. (1) P4, L.23: 875 PgC. This is another result and should therefore be in the result section (2) done

18. (1) The results section needs to be rewritten completely with a focus on your own results. (2) done

19. (1) P5, L.5: Add the year of publication after Yu (2) done

20. (1) P5, L.10: Add the year of publication after Yu (2) done

21. (1) P5, L.10: Please change "one could find" into "it is reasonable to agree: : :" (2) done

22. (1) Please provide a more in-depth discussions of your methodological approach, e.g. show the benefits but also limitations of your model and compare your results of potential C accumulation with e.g. C accumulation during the Holocene. I suggest making several subchapters. One where you discuss the benefits and limitations of your model, including the uncertainty of your estimation. Another subchapter where you compare your results with previous studies (as you did in the results section) and a third subchapter where you discuss the implications of your results on the global C cycle (basically your actual discussion).

(2)The uncertainty of estimates is explained in the methods section (sub-section 2.3), because we apply an original method for characterizing the uncertainty of our estimates.

(3) The section Discussion is rewritten.

23. (1) P5, L.14: Change the first sentence into: "The potential for northern peatlands to store carbon were estimated for the first time: : :"

(2) Yes, "potential for growth" and "limits to growth" have the same meaning, and "potential for growth" sound more positive. However, we would like to keep here a "connotation" to the paper "The limits to peat bog growth" by Clymo (1984) cited in the next phrase.

24. (1) P5, L.15f: Change the following sentence into: "We adapted this methodology to global scale and additionally included geomorphological aspects of peat bog growth: : :"

(2) We changed this sentence proceeding from general idea of this recommendation.

(3) We adapted this methodology for use at the Earth system scale based on the gridded data (Hengl et al., 2014) representing geomorphological aspects of peat bog growth.

25. (1) P5, L.18 : Write "Our estimate: : :" instead of "Moreover, this: : :"

(2) We deleted "Moreover".

(3) "This estimate, 875±125 PgC, corresponds to the present climate. . ."

26. (1) P5, L.18: Delete "somewhat"

(2) done

27. (1) P5, L.19: Change the following sentence into: "This assumption, however, might not be relevant for scenarios of dramatic changes in the Earth system that will

take place if cumulative carbon: : :"

(2) done

28. (1) P5, L.21f: Change the following sentence into: "Nevertheless, if cumulative carbon emissions do not exceed 1000 PgC, the northern peatlands play an important role in global carbon cycle recovery"

(2) done

29. (1) P5, L.26: Replace "plain" with "other" (2) done

30. (1) P6, L.1-4: You should also discuss the conditions and timeframe under which such a scenario can happen. Also, if you make such a strong statement, there should be a better explanation of this Earth system model of intermediate complexity.

(2) It is not a statement; it is rather a report about the numerical experiment that demonstrates the role of northern peatlands in global carbon cycle recovery and calls for further numerical experiments. That is why it is presented in discussion. The Earth system model of intermediate complexity, CLIMBER-2, is described in the cited paper.

31. (1) P6, L.3 replace "won't" with "will not be able to" (2) Done

32. (1) Please add a reference list for the supplement (2) done

33. (1) S1.1: Please rephrase the first sentence. (2) done

---

## Author Comment (AC2) · 18 Jun 2019

We are pleased to see that Referees agrees that the results of the study are worth publishing. We also acknowledge the importance of Referee's comments to strengthen the presentation of the results. Below is our response to the Referee's questions and recommendations.

Response to the questions asked by Referee:

1. (1) P1 L18: How did you define northern peatlands (> 40N or 45 N)?

(2) Here we refer to the article of Loisel et al. (2017) and keep in mind the peatlands located north of 45 N.

[Figure]

(3)". . . northern peatlands, namely the peatlands distributed across the northern mid- and high-latitude regions located north of 45°N, . . ."

2. (1) P1 L19: "The variations are explained by. . ." Which variations?

(2) Here we keep in mind the variation in the carbon sink magnitude mentioned in the previous sentence (P1 L19).

(3) "The variations in the sink magnitude. . ."

3. (1) P1 L22: "However, during the last 5000 years, the area of peatlands remained relatively stable . . ." Peat basal ages are used as proxies to identify new peatland areas and expansion rate. From figures 1 and 3 in MacDonald et al. 2006, we can see that around 30-40

(2) Here we cited the estimates of Bog/Swamp area given in the "Table 1. Reconstructed surface area of ecosystems" in the Adams and Faure (1998) article: 1.85 Mkm2 by 8000 BP, 2.35 Mkm2 by 5000 BP, 2.45 Mkm2 by present. The cumulative curve of 1516 radiocarbon dates of basal peat deposits shown at the at the Figure 3 in the article of MacDonald et al. (2006) also show that major part (70 percents) of the studied peatlands was initiated before 5000 BP. Moreover, MacDonald et al. (2006) wrote, "new peatland initiation was relatively modest in the late Holocene", that seemingly had the same meaning as the phrase "the area of peatlands remained relatively stable in the late Holocene", and led to conclusion that the growth in the peat depth was responsible for the major part of the carbon uptake in the late Holocene, whereas peatland expansion was responsible for the minor part of the carbon uptake in the late Holocene.

(3) Taking into account considerable uncertainty over peatland area at any particular time of the past, we agree that it would be more correct to say, "Since the area of peatlands remained relatively stable in the late Holocene, the major part of the carbon sink provided by northern peatlands during this period could be attributed to the growth

in peat depth, not to the growth of the area occupied by the northern peatlands".

4. (1) P1, L.25: "the northern peatlands may accumulate 864-2200 PgC . . . " This is a very high value, how did you calculate this range. From where did you find this information? What about the peatland distribution area and sink capacity, will they remain the same in the future? Studies indicated that many peatlands would lose their carbon sink capacity while some may enhance

(2) This range was calculated as follows. "The average rate of carbon accumulation associated with peat growth is estimated at 18-28 gC m-2 yr-1 (Yu, 2001)" (P1, L.24). Northern peatlands occupy 2.4-4 million km2 (Yu, 2011) (P1, L.25). Hence, during the 20000 $(2*10^4)$ years, the northern peatlands may accumulate from $(18*2.4*10^{12})*(2*10^4) = 86.4*10^{16} = 864*10^{15}$ gC to $(28*4*10^{12})*(2*10^4) = 224*10^{16} = 2240*10^{15}$ gC. This is an estimate of cumulative carbon uptake that could be provided by peatlands under the present peatland area and the present average carbon accumulation rate. Our research, in fact, is based on the hypothesis that average carbon rate must decline in the future.

(3) "This rate suggests that northern peatlands, occupying 2.4-4 million km2 (Yu, 2011), may accumulate during the next 20,000 years the amount of carbon comparable to the expected cumulative anthropogenic carbon emissions corresponding to a 2.5oC warming (Raupach et al., 2014), namely from 864 PgC (18 gC m-2 yr-1 *$2.4*10^{12}$ m2 * $2*10^4$ yr) to 2240 PgC (28 gC m-2 yr-1 * $4*10^{12}$ m2 * $2*10^4$ yr)." We also move the next paragraph to the end of Introduction, to start the discussion of changes in the rate of carbon accumulation immediately after these estimates.

5. (1) P2 L6: How did you estimate this range – see my previous comment. (2) See the response to the question No. 4.

6. (1) P2 L11: "at least a small portion of the organic matter that enters the acrotelm always reaches to the catotelm . . ." Is this a plausible argument – do you think, acrotelm always passes organic matter in the catotelm? Even when peatland experiences continuous dry conditions?

(2) Here we keep in mind an accumulating peatland, not a degrading peatland: "at least a small portion of the organic matter that enters the acrotelm always reaches the catotelm in an accumulating peatland".

(3) We may put more accent to the context of this phrase by adding the sentence, "This is, of course, not true in the case of a degrading peatland, but degrading peatlands do not fall within the scope of this study".

7. (1) P2 L 13-15: In which study, did you find this information?

(2) This conceptual scheme summarizes and generalizes a number of studies, but the closest source is the article of Alexandrov, Brovkin, and Kleinen (Sci. Rep., 6, doi:10.1038/srep24784, 2016)

(3) The maximum height of the water table, and thus the potential peat depth, is determined by the amount of effective rainfall, drainage system density and the hydraulic conductivity of peat and mineral materials below the peat (Alexandrov et al., 2016).

8. (1) P2 L27: How did you determine where to form a cluster in a grid cell?

(2) The location of a peatland cluster in a grid cell does not affect the estimate of the potential amount of carbon that could be accumulated in the grid cell, therefore it is not determined.

(3) To avoid possible misunderstanding, we change "clustered distribution "to "non-uniform distribution", "The conservative estimate assumes uniform distribution of peatlands over all grid cells ($f_{PW} = f_{P,obs}$; $f_{WP} = 1$), the non-conservative estimate assumes non-uniform distribution over all grid cells ($f_{PW} = 0.75$; $f_{WP} = f_{P,obs}/0.75$), and the less-conservative estimate is derived using a rule-based algorithm categorizing the grid cells into those where peatland distribution is uniform and those where peatland distribution is non-uniform."

9. (1) P3, L.3: What is the density of draining system?

(2) The density of draining system is the length of draining streams per unit area.

(3) ". . .the potential peat depth, is determined by the amount of effective rainfall, drainage system density (the length of draining streams per unit area) and the hydraulic conductivity . . ."

10. (1) P3 L4: "The impeded drainage model approach" – Give more details about this approach and model. What it is and where this approach has been used before?

(2) The details of the impeded drainage model approach are given in the Supplement (https://www.biogeosciences-discuss.net/bg-2019-76/bg-2019-76-supplement.pdf). The basic idea of the approach is that the maximum peat depth is equal to the maximum height of water table above the level of draining system calculated using the impeded drainage model plus the maximum depth of acrotelm.

(3) "To calculate the potential peat depth, we apply an equation derived (see Supplement) from the impeded drainage model used in our previous study (Alexandrov et al., 2016)?.".

11. (1) P3 eqn 1 - From where this equation comes from? Any previous applications?

(2) This equation was derived from the equations of the impeded drainage model, see equations (S1-S17) in the Supplement (https://www.biogeosciences-discuss.net/bg-2019-76/bg-2019-76-supplement.pdf).

(3) This part of the text is rewritten.

12. (1) P3 L10: There are many peatlands in the southern latitude region between 45-55N, particularly in China, U.S and Mongolia. Have you considered them in your calculation?

(2) Yes, we considered the peatlands located north of 45 N.

(3) " . . . sink provided by northern peatlands, namely the peatlands distributed across the northern mid- and high-latitude regions located north of 45°N, . . ."

13. (1) P3 L18: Did you check the recent study by Xu et al. 2018 where the authors have refined the global and regional estimates of peatland distribution area? How your dataset (WISE30sec) is different or better than Xu et al. 2018 (PEATMAP)?

(2) The WISE30sec data set (Batjes, 2016) of soil properties is based on Harmonised World Soil Database (HWSD). The differences in the estimates of peatland area between HWSD and PEATMAP are reported in the Table 2 of the article published by Xu et al (2018): 1.327 vs 1,339 Mkm2 for North America, 0. 879 vs 1.180 Mkm2 for Asian Russia, 0.634 vs 0.528 Mkm2 for Europe. It does not seem that these differences may dramatically affect our conclusion that it might be reasonable to agree that the estimate of $875 \pm 125$ PgC, as obtained from two completely independent methods, is the most expedient estimate of the potential carbon stocks in northern peatlands. At the same time, we agree that it is important to trace the effect of input data updates. Therefore, we are going to publish the source code of the computer programs that were used in calculations. This source code could be employed by anyone for updating our estimate in response to the updated information on peatland area.

(3) "Analyzing the uncertainty in the data on present-day peatland extent goes beyond the scope of this study. Improving the accuracy of these data is a well known task tackled by ISRIC, the International Soil Reference and Information Centre, (Batjes, 2016; Hengl et al., 2014), and by networks of peatland scientists such C-Peat (Treat et al., 2019) and PeatDataHub (Xu et al., 2018). Hence, it might be more important to update the estimates of potential carbon stocks on a regular basis to keep pace with improvements in the accuracy of the data on present-day peatland extent."

14. (1) P4 L: How accurate are these conservative and non-conservative estimates? From Table 1, one can see that both estimates fail to capture the observed peatland carbon density. In fact, in some cases, the conservative estimates are higher than the

observed values. Based on this information, do you think we can rely on your modelled limits?

(2) Both the conservative and non-conservative estimates are not the estimates of the present peat carbon density: they are estimates of the maximum peat carbon density that could be achieved in the future, under given climatic and geomorphological conditions. Therefore, they should not capture the observed peatland carbon density. The fact that the non-conservative estimates are significantly higher than the observed carbon densities allows the following interpretation: the sites listed in the Table 1 are far from equilibrium and could accumulate a large amount of carbon by the end of the current interglacial. As to the conservative estimates, which are lower than the actual peat carbon density at the sites that fall within the grid cells where $f_{P,obs}$ is less than 20

(3) This part of text is rewritten.

15. (1) P5 L15: There are other methodologies which have been developed to estimate total carbon stocks (see Yu et al. 2012). How your approach is different or better than these methodologies and what are its limitations?

(2) These methodologies are to estimate present carbon stocks. We estimate the carbon stocks that could be achieved in the future. Our approach for estimating the future carbon stocks is similar to peat volume approach, but the estimate of the mean peat depth in a given region is replaced by the estimate of the maximum mean peat depth that could be achieved in the given region. We also compare our estimate of the future carbon stocks with those we derived from the time history approach.

(3) "The results of our study suggest that even the conservative estimate of the potential carbon stocks (665 PgC) is still higher than Gorham's (1991) estimate of 455 PgC in the actual carbon stocks of northern peatlands. Gorham's estimate, based on peat-volume approach, . . ." . . . "The conservative estimate is also higher than the Yu's (2011) estimate of actual carbon stocks, $547 \pm 74$ PgC, based on the time history approach,

suggesting that northern peatlands in total would accumulate in the future more carbon than they store now.

16. (1) P5 L16: "We adapted this methodology for use at the global scale ..." Global or regional because you have considered only the northern peatlands?

(2) It may be more correct to say, that we adapted this methodology for use in the studies of the Earth climate system, as we found that northern peatlands are the important element of the Earth climate system affecting the length of the current interglacial.

(3) "We adapted this methodology for use at the Earth system scale ..."

17. (1) P6 L7: "If there were no limits to their growth ..." In the introduction, you have mentioned that peatlands can reach to steady state and do not grow or accumulate carbon after that. Do your analysis shows in which regions peatlands have already reached to steady state?

(2) If there would be a map of carbon stocks in peatlands, the comparison of this map to the map displayed at the Fig.3 (the less-conservative estimate of the potential carbon stocks) would show in which regions peatlands have already reached to steady state. At the moment, only the grid cells where $p_{C,max}/(A * f_{P,obs}) \leq 45$ KgC m-2 could be categorized as the grid cells where peatlands already reached the steady state.

Response to recommendations made by Referee:

1. (1) P2 L 1-15: Support your arguments with previously established knowledge. Include references. (2) Done

2. (1) P2 L5: Define what a steady state is for your readers. (2) Done (3) " ... the closer the peatland ecosystem is to its steady state, that is, to the equilibrium between organic matter production and decomposition, the lower is the carbon sink magnitude."

3. (1) P2 L9: Remove this expression – "the so called" (2) Done

4. (1) P2 L 16: Could you explain a bit about your model. What it does and other

relevant information briefly and give more details in the methods section. (2) Done (3) This part of the text is rewritten.

5. (1) P2 L 22: "The gridded data on soil properties give the fraction of a grid cell covered by peatlands : : :" Include reference. (2) done

6. (1) P2: I think a paragraph needs to be added in the end which explains the purpose of your study. (2) done

7. (1) Methods: You could start with an equation for the maximum depth of peat before introducing the maximum carbon stock in a grid cell (2) Done

8. (1) Methods. Perhaps subheadings could be helpful to improve and clarify the structure of the methods. I also suggest you to add a model description section.

(2) done

(3) We divided the Methods section into: "Equations", "Input data", and "Uncertainty associated with peatlands distribution over a grid cell"

9. (1) P3 L7-8: it is better to include the value of constants in the equation or under it. (2) done

10. (1) P3 L11: Include a brief write up about the SoilGrid dataset and what it contains.

(2) We use only the depth to bedrock from this dataset and give reference to the paper where this data set is described in detail.

11. (1) P3 L22: If you have the dataset then you can easily estimate how much area is occupied by northern peatlands. According to Xu et al., around 3.12 million km2 area is occupied by peatlands above 45N and Yu et al. 2010, used 4.0 million km2.

(2) Yes. We corrected this phrase.

(3) "These data allow us to estimate the values that fP may take at the cells of the 0.1°*0.1° geographic grid (Figure 2) and the total area, $2.86 * 10^6 km^2$, that peatlands

occupy in the land north of 45N."

12. (1) P4 Results: This looks like a part of the discussion. I recommend you to explain your results and what you see in your figures before comparing them with the previously established knowledge.

(2) done

13. (1) P4 L27: You can also include the eqn used by Gorham 1991- Cpeat = Pi (Ai $\times Di \times BDi \times CCi$)

(2) We did not find this equation in the cited Gorham's paper. Therefore, we supposed that it might be better to explain the Gorham's version of the peat-volume approach through an indirect quotation of his words. Here is the direct quotation, "... we can then estimate readily the total carbon in the dry mass of boreal and subarctic peat, subtracting the mined area, as $(3.42 \times 10^{12} m^2) \times (2.3m) \times (112 \times 10^3 g/m^3) \times (0.517) = 455 \times 10^{15} g$, or 455 petagrams (Pg)."

14. (1) P4 L29: $112 \times 10^3$ g m-3 - Change it to 112 kg m-3

(2) This is a part of an indirect quotation; therefore, we suppose that it might be better to keep the units in the same format as they were in Gorham's paper.

15. (1) P5 L4: Explain what time history approach is.

(2) Since it is difficult to explain this approach in few words, we suppose that it might be better to give a reference to the publication, where this approach is explained in detail.

16. (1) P5 L18-22: No references.

(2) References are inserted.

---

## Author Response (AR1)

**The limits to northern peatland carbon stocks**

 by G. A. Alexandrov, V. A. Brovkin, T. Kleinen, and Z. Yu

*Biogeosciences Discuss.*,

https://www.biogeosciences-discuss.net/bg-2019-76/#discussion

Point-by-point reply to all comments

All comments were considered thoroughly, and most of them were addressed in the revised manuscript. The following comments are not addressed in the manuscript: the comments #34, #41, #43, #44, #47 and #47 of Reviewer#1 and the comments #29 and #35 of Reviewer #2. We are looking to receiving Editor's advice on how to address these comments.

The help or reviewers is acknowledged in the revised manuscript.

**Point-by-point reply to Editor's comments**

| No | Comment | Reply | Changes in the manuscript |
|---|---|---|---|
| 1. | In your response letter you should address each review individually. Your response letters, i.e. your author comments, are a mix of both reviews. This way it is impossible to evaluate by me if the requests by the reviewers have been addressed adequately. I think your author comment to reviewer 1 is missing, please provide a separate author response to reviewer 1. | In this point-by-point reply, each review is addressed individually. We also checked the posted author replies. The author's reply to reviewer #1 comments, AC1, is posted here: https://editor.copernicus.org/index.php/bg-2019-76-AC1.pdf?_mdl=msover_md&_jrl=11&_lcm=oc108lcm109w&_acm=get_comm_file&_ms=75014&c=163220&salt=1695120521 A separate author response to reviewer 1 is also provided in | N/A |

| | | this point-by-point reply (p. 3 - 16) | |
|---|---|---|---|
| 2. | You need to address the reviewer concern (reviewer comment 1) described in the general comment on the methods (page C3), results (page C4) and discussion (page C5) very seriously. Please provide an in-depth description in your revised author comment on how you have addressed the description of the methods, the critique on how to present the results and the discussion in your manuscript. | All reviewer concerns were considered seriously. We rewrote the sections "Methods", "Results" and "Discussion" to address reviewer comments. In depth description of the changes can be found at pages of this point-by-point reply. | N/A |
| 3. | Your response to reviewer 2, point 10. (1) P5, L.20-21: Why 1000 PgC? It seems a bit arbitrary to me." Please consider latest estimations on the anthropogenic carbon budget to achieve the 2-degree climate target, which has been refined to 1.5°C. Please refer to the 1.5degree special report and references therein and revise your calculated number for the carbon budget and the references used for justification accordingly. | Thank you for advice to use IPCC 1.5 report to justify the range of validity for the estimate of the potential carbon stocks in northern peatlands. Indeed, the sum of historical cumulative emissions and the future cumulative emissions compatible compatible with the global average temperature increase to below 2°C reported in the Chapter 2 gives 1000 PgC. | "The recent analysis of mitigation pathways compatible with global warming of 1.5°C above pre-industrial levels (Rogelj et al., 2018) shows that holding the global average temperature increase to well below 2°C is difficult but not impossible. To achieve this goal, cumulative $CO_2$ emissions from the start of 2018 until the time of net zero global emissions must be kept well below 1430 $GtCO_2$, (i.e., 390 PgC), that corresponds to 66th percentile of transient climate response to cumulative carbon emissions (Rogelj et al., 2018; Table 2.2). Since cumulative $CO_2$ emissions through to year 2017 are estimated at 610 PgC (Le Quéré et al., 2018), 1000 PgC of cumulative carbon emissions, the sum of historical (610 PgC) and the future cumulative emissions compatible with the global average temperature |

| | | | increase to below 2°C (390 PgC) could be considered as a threshold for defining the range of validity of the most expedient estimate of potential carbon stocks in northern peatlands. " – P6, L29-P7,L6 |
|---|---|---|---|
| 4. | Please check carefully if you have not missed individual comments in your author comment and provide the missing answers. Please note that you have to address each comment individually, therefore it is important to provide point-by-point author comments to each review. | The point-by-point author comments to each review is provided below. To be sure that no individual comment is missed, point-by-point reply is provided to the text of each reviewer comment. The text of each comment is divided to logical parts cited in the second column of the Table, reply to this part is provided in the third column of the Table, and changes in text are described in the fourth column of the Table. | N/A |

**Point-by-point reply to Reviewer #1 comments**

| No | Comment | Reply | Changes in the manuscript |
|---|---|---|---|
| 1. | Alexandrov et al. raise an interesting topic and modeled the potential for carbon sequestration in northern peatlands. They show that large amounts of carbon in the atmosphere could be offset by peatland growth throughout the current interglacial. I think the study focuses on an important topic and the results are worth publishing, however, the methods and the results need to be presented in a revised, more precise and coherent form. I think the paper | The methods and the results are presented in more precise and coherent form in the revised manuscript. | The sections "Methods" and "Results" are re-written to address reviewer's questions and recommendations. |

| | | | |
|---|---|---|---|
| | should be significantly revised before consideration of publication. Please see my detailed comments below. | | |
| 2. | Abstract:
Please insert one or two statements about the methods, which you applied in this study. Also, include a statement about your results, where you specifically mention the amount of carbon which could be set off by peatland growth. | We explained that our results were derived from the gridded data on the depth to bedrock and on the fraction of area covered by soils of histosol type and mentioned that 330±200 PgC is the amount of carbon that could be set off by peatland growth. | "The limits to the growth of northern peatland carbon stocks, evaluated based on the gridded data on the depth to bedrock and on the fraction of area covered by soils of histosol type, suggest that 875±125 PgC is the most expedient estimate of the potential carbon stock in northern peatlands at large and that 330±200 PgC is the most expedient estimate of the total amount of carbon that they could remove from the atmosphere during the period from present to the end of the current interglacial." – P1, L12-l6 |
| 3. | In addition, I would recommend changing the title of the manuscript into "The potential of northern peatlands for carbon sequestration" | We would like to keep a "connotation" to the paper "The limits to peat bog growth" by Clymo (1984), and to highlight the fact that the cumulative amount of carbon that northern peatlands could remove from the atmosphere is limited by the geomorphological conditions in present climate. | Based on the overall idea of the suggested title, we think it would be reasonable to change the title as follows, "The limits to growth of northern peatland carbon stocks", if Editor does not mind. |
| 4. | Specific comments:
Page 1, Line 10: Maybe write "continuous" instead of "persistent" | We used "persistent" instead of "continuous", because "persistent carbon sink" is a common collocation appeared in a number of research articles on carbon cycle (e.g., Pan, Y. D. et al. A large and persistent carbon sink in the world's forests. Science 333, 988–993 (2011)). | We did not make changes in response to this recommendation, but we can do this if necessary. |
| 5. | P1, L.12: Rewrite the sentence. E.g. "The evaluation of the carbon | Here we were trying to say that over the next 5 thousand years | "This leads to conclusion that northern peatlands, not only the |

| | | | |
|---|---|---|---|
| | sequestration potential of northern peatlands show that atmospheric carbon dioxide concentration can be significantly reduced. Northern peatlands have the potential to be the second largest CO2 sink after the world's oceans." | after the end of fossil fuel burning, not only oceans but also northern peatlands will be removing carbon dioxide from the atmosphere. | oceans, will potentially play an important role in reducing the atmospheric carbon dioxide concentration over the next five thousand years." – P1, L16-18 |
| 6. | Introduction: General comments: The introduction needs a better structure. The different paragraphs need to be connected better and the research gap should be mentioned more clearly. | Done | We moved the paragraph that seemingly was breaking the logic flow to the proper place in the end of Introduction. –P2, 23-30. |
| 7. | The last two paragraphs (p2, line 17-29) belong into the methods part and should be removed from the introduction | Done. | We removed these paragraphs from Introduction. |
| 8. | Specific comments: Page 1, Line 17: You mention the study by Loisel et al. (2014). Please also include the new study by Treat et al. (2019) in your introduction | Done. | "The recent compilations of peatland data (Loisel et al., 2014; Treat et al., 2019) largely confirm …. "– P1, L20 |
| 9. | P1, L.17 : I suggest to use the word "knowledge" instead of "wisdom" | We think that "notion" would be good. | "… largely confirm the conventional notion of the carbon (C) sink provided by northern peatlands …" |
| 10. | P1, L.21 : I suggest to use the word "previous" instead of "later" | Here we use 'later' in sense of 'coming after something else'. Hence, it cannot be replaced by "previous". | To avoid possible misunderstanding, we revise this sentence as follows, "In the early Holocene, both the rate of peatland expansion and the rate of carbon accumulation appear to be highest (Yu et al., 2010) as compared to the later Holocene periods." – P1, L23-26 |
| 11. | P1, L.23 : Where do northern peatlands start? Is it >40 North or >45 North, please clarify | Here we refer to the article of Loisel et al. (2017) and keep in mind the peatlands located north of 45 N. | "… northern peatlands, namely the peatlands distributed across the northern mid- and high-latitude regions located north of 45°N, …" – P1, L21-22 |

| 12. | P1, L.25: 864-2240 PgC – Is that already your result or is it from a different study – please clarify | These numbers were calculated from the range of estimates of carbon accumulation rates associated with peat growth and the range of estimates of peatland area reported by Yu (2011) and cited in this paragraph. It is a starting point of our research aimed to explore limitations to peatlands growth that do not allow them to remove 2000 PgC amount of carbon from the atmosphere. The changes made in the manuscript hopefully make it clear that this is a simple extrapolation based on the estimates reported by Yu (2011). | "This rate suggests that northern peatlands, occupying 2.4-4 million $km^2$ (Yu, 2011), may accumulate during the next 20,000 years the amount of carbon comparable to the expected cumulative anthropogenic carbon emissions corresponding to a 2.5°C warming (Raupach et al., 2014), namely from 864 (18 gC $m^{-2}$ $yr^{-1}$ ×2.4·$10^{12}$ $m^2$ × 2·$10^4$ yr) to 2240 (28 gC $m^{-2}$ $yr^{-1}$ × 4·$10^{12}$ $m^2$ × 2·$10^4$ yr) PgC." – P2, L1-5 |
|---|---|---|---|
| 13. | P2, L.13: I suggest to use the word "rise" instead of "elevation" | Done. | "The rise of groundwater is caused by the rise of the peatland surface that in turn results from accumulation of organic matter." – P2, L18-19 |
| 14. | Methods: General comments: The methods are somewhat unclear to me. You start with an explanation of the maximum depth of peat, however in equation 1 you show how the maximum C stock can be calculated. You could start with an equation for the maximum depth of peat before introducing the maximum carbon stock in a grid cell. | Done. | "To calculate the potential peat depth, we apply an equation derived (see Supplement) from the impeded drainage model used in our previous study (Alexandrov et al., 2016). This equation …" – P3, L3-8 |
| 15. | In addition, I suggest to make subchapters to explain the different model parameters. The first subchapter could include the maximum carbon stock in a grid cell, whereas a second subchapter includes the extrapolation from the grid cell to the entire northern | We subdivided "Methods" into "Model equations", "Input data", and "Uncertainty associated with peatlands distribution over a grid cell". The latter subsection is to explain the difference between the conservative and non-conservative estimation of $f_p$. | The estimate for the entire peatland area is merely a sum carbon stocks in the grid cells located north of 45N, and hence extrapolation from the grid cell to the entire northern peatland area is explained in one sentence: "The sum of the |

| | | | |
|---|---|---|---|
| | peatland area and a third subchapter explains the differences between a conservative and non-conservative interpretation of fp. | Maximum carbon stock in the grid cell is explained in the first section. | potential carbon stocks for all cells north of 45ºN gives the conductivity-dependent estimate of the potential carbon stock in northern peatlands." – P5, L5-7. |
| 16. | Also, in the end of the methods, it appears to be a mix of discussing your methods and presenting some results already. I suggest you discuss your methods in the discussion section with a separate subchapter and strictly separate between methods and results, so that no results appear in the methods section. | Done. | No results appear in the methods section. The discussion section is rewritten. |
| 17. | Specific comments: P3, L.3: What is the density of draining system – please explain | The density of draining system is the length of draining streams per unit area. | "…the potential peat depth, is determined by the amount of effective rainfall, drainage system density (the length of draining streams per unit area) and the hydraulic conductivity …" – P2, L21 |
| 18. | P3, L.4: What is the impeded drainage model? – If this is your own model, you should explain it in the methods, otherwise add a reference. | The impeded drainage model is the model based on the Dupuit-Forchheimer theory of groundwater movement (aka hydraulic theory) and a few additional assumptions (see Supplementary Information to our previous work, https://media.nature.com/original/nature-assets/srep/2016/160420/srep24784/extref/srep24784-s1.doc). The basic idea of the model is that the high level of water table in a peatland is maintained due to impeded drainage: it takes long time for water coming with precipitation at the central part | "To calculate the potential peat depth, we apply an equation derived (see Supplement) from the impeded drainage model used in our previous study (Alexandrov et al., 2016)." – P3, L3-4 |

| | | of a peatland to reach the draining streams. | |
|---|---|---|---|
| 19. | P3, L.6: I do not understand the second, smaller equation. Why is hmax, the maximum height of the water table above the level of the draining system, dependent from the fraction of the area occupied by peatlands? | Both $h_{max}$ and $f_p$, the fraction of the area occupied by peatlands, depend on $K$, the hydraulic conductivity: equation (S6) and equation (S10) in the Supplement (https://www.biogeosciences-discuss.net/bg-2019-76/bg-2019-76-supplement.pdf). The "observed" value the fraction of the area occupied by peatlands, $f_{P,obs}$ makes it possible to estimate $K$: equation (S11). Substituting K given by equation (S11) to equation (S6) gives the equation (S12), where $h_{max}$ depends on $f_{P,obs}$ and g, the average height of the watershed above the level of the draining system. That is to say, excluding K from the equation for $h_{max}$ leads to including $f_{P,obs}$ into this equation. | This part of the text is rewritten, and hopefully the phrase "To calculate the potential peat depth, we apply an equation derived (see Supplement) from the impeded drainage model used in our previous study (Alexandrov et al., 2016). This equation relates the maximum height of the water table above the level of draining system, $h_{max}$, at a given watershed to the fraction of its area covered by peatland, $f_{P,obs}$, and the average depth to bedrock, $g$ … " – P3, L3-6 – makes it clear that a detailed explanation could be found either in Supplement or in the previous publication. |
| 20. | P3, L.9: Change the sentence to ": : :hmax is the maximum height of the water table above the level of the draining system: : :" | Done. | "This equation relates the maximum height of the water table above the level of the draining system, $h_{max}$, at a given watershed to  .." – P3, L3-5 |
| 21. | P3, L.26: How much is the minimal depth of the peat layer which is used to classify a land unit as peatland? – Please give a number or a range for the minimal peat depth. | The minimal depth of the peat layer which is used to classify a land unit as peatland is a source of uncertainty in the estimates of peatland area. We relied on the WISE30sec data set (Batjes, 2016) of soil properties and diagnosed peatland extent by fraction of grid cell covered by soils of histosol type. Hence, the minimal depth of the peat layer is assumed to be 40 cm | "The estimates of the actual peatland area may vary depending on the criteria that are used to distinguish peatlands from other types of land surface. The minimal depth of the peat layer, which is used to classify a land unit as peatland, is the criterion that affects the estimates of peatland area (Xu et al., 2018) . Since peatland extent is diagnosed by the extent of |

| | | | histosols, $2.86 \times 10^6$ km$^2$ should be interpreted as an estimate of the area of peatlands with peat depth exceeding 40 cm (according to FAO definition of histosols)." – P4, L2-6 |
|---|---|---|---|
| | | (according to FAO definition of histosols) | |
| 22. | P4, L.1-11: This part would better fit into the discussion where you could have a subchapter discussing your methods and you model. | This part is an explanation of our approach to addressing uncertainty. We rewrote the text to clarify this point. | "**2.3 Uncertainty associated with peatlands distribution over a grid cell**

The gridded data on soil properties give the fraction of a grid cell covered by peatlands. To estimate the fraction of a watershed covered by peatlands, $f_{PW}$, which is needed for calculating $h_{max}$, one should make an assumption about the peatland distribution within the grid cell. …" – P4, L7 - 18 |
| 23. | P4, L.12: What is the non-conservative and what is the conservative interpretation of fp,obs, please add values | The variety of possible interpretations of $f_{p,obs}$ is parameterized using the equation (S18) in the Supplement. The difference between the conservative and non-conservative interpretations of $f_{p,obs}$ could be illustrated by the following example. Let us consider a grid cell the 36% of which is covered by peatlands. Does it mean that peatlands cover 36% of each watershed within this grid cell? Or does it mean that only 48% of watersheds are occupied by peatlands, and the peatlands cover 75% (0.48*75=36) of each of these watersheds? In other words, we cannot say for sure whether the grid cell contains many small peatlands, or few large peatlands. Under the | In the revised version of the manuscript this part of the text is re-written. We change wordings: 'conservative interpretation' to '**uniform interpretation**', and 'non-conservative interpretation' to '**clumped interpretation**'.

"To estimate the fraction of a watershed covered by peatlands, $f_{PW}$, which is needed for calculating $h_{max}$, one should make an assumption about the peatland distribution within the grid cell. …The uniform estimate assumes a uniform distribution of peatlands over all grid cells ($f_{PW}= f_{P,obs}$; $f_{WP}=1$), the clumped estimate assumes a non-uniform distribution over all grid cells ($f_{PW}=0.75$; $f_{WP}=f_{P,obs}/0.75$) … As it can be seen from Table 1, the estimates |

| | | | conservative interpretation, $f_{p,obs}$ = 36% suggests that peatlands cover 36% of each watershed within this grid cell (many small peatlands). Under the non-conservative interpretation, $f_{p,obs}$ = 36% suggests that only 48% of watersheds within the grid cell are occupied by peatlands, and the peatlands cover 75% (0.48*75=36) of each of these watersheds (few large peatlands). The conservative interpretation of $f_{p,obs}$ leads to smaller estimate of pmax as compared to the non-conservative interpretation. | of the potential peat carbon density based on the uniform interpretation of $f_{P,obs}$ ($f_{PW}=f_{P,obs}$; $f_{WP}=1$) are often lower than the actual peat carbon density at the sites that fall within the cells where $f_{P,obs}$ is low." – P4, L15-17. |
|---|---|---|---|
| 24. | P4, L.13: 1258 vs 665 PgC. This is a result and should therefore be in the results chapter | Done. | "The full range of uncertainty for the estimate of the amount of carbon that northern peatlands may accumulate from the start to the end of the current interglacial could be characterised by the uniform and clumped estimates. The former is equal to 665 PgC, and the latter is equal to 1258 PgC." – P5, L21-23 |
| 25. | P4, L.14: Please replace "one cannot expect: : :" with "it cannot be expected: : :" | Done. | "This uncertainty cannot be easily reduced by using a finer grid, because it cannot be expected that each watershed falls within one grid cell." – P6, L4-5 |
| 26. | P4, L.18: Please replace "one may assume: : :" with "it can be assumed: : :" | Done. | "If K is above the typical value, $K_c$, then it can be assumed that peatland occupy $f_{p.obs}$ / $f_{p.est}$ fraction of watersheds and cover $f_{p.est}$ fraction of area of each of these watersheds, where $f_{p.est}$ is set at the value that brings K to $K_c$." – P4, L21-23 |

| 26. | P4, L.21: "peat C addition" do you mean C accumulation? | We use the words "peat C addition" to denote the amount of carbon that enter to catotelm. Peat accumulation is the difference between peat addition and peat decomposition. | The words "peat C addition" are changed to "annual C input to catotelm': "This model suggests that the growth of carbon stock in peatlands is limited by the ratio of annual C input to catotelm to the decay constant." – P4, L26-27 |
|---|---|---|---|
| 27. | P4, L.23: 875 PgC. This is another result and should therefore be in the result section. | Done. | "The sum of the potential carbon stocks for all cells north of 45ºN gives the conductivity-dependent estimate of the potential carbon stock in northern peatlands, which is " – P5, L15-17 |
| 28. | Results
General comments:
Please present here your own results and do not start with a comparison to another study. Instead of all the numbers from Gorham (1991), present your own results for mean depth of peatlands, mean bulk density or area of peatlands. The comparison with Gorham (1991) as well as Yu (2011) belongs to the discussion part. The results section needs to be rewritten completely with a focus on your own results. | Done. | We re-wrote the section "Results" completely to focus on our own results and move all comparisons to the 'prior art' to "Discussion". |
| 29. | Specific comments:
P5, L.5: Add the year of publication after Yu | Done. | This sentence was completely re-written in the revised version. |
| 30. | P5, L.10: Add the year of publication after Yu | Done. | "The clumped estimate, 1258 PgC, is beyond the range of uncertainty, 760-1006 PgC, in the estimate of potential carbon stocks that could be derived using the Yu's (2011) model of peat accumulation (see Supplement)." – P6, L21-22 |

| 31. | P5, L.10: Please change "one could find" into "it is reasonable to agree: : :" | Done. | "Hence, it is reasonable to agree that the estimate of 875±125 PgC, as obtained from two completely independent methods, is the most expedient estimate of potential carbon stocks in northern peatlands …" – P6, L22-23 |
|---|---|---|---|
| 32. | P5, L.11: Why 875 PgC? What is with the 665 PgC – 1258 PgC? What is your main result? This needs to be clear. | The main result of this study is the expedient estimate of carbon stock that could be accumulated by northern peatlands by the end of the current interglacial. This estimate is equal to 875 PgC and falls within the range of uncertainty that starts from 750 PgC to 900 PgC (= 875±125 PgC), and derived from the Yu's model (Yu, 2011). The validity of this estimate is supported by the estimates of potential carbon stocks obtained by a completely independent method under different interpretations of the data on the georgraphic distribution of peatlands: this estimate, 875±125 PgC, falls within the range of uncertainty associated with accuracy of the data on the georgraphic distribution of peatlands. | This part of the text was re-written: "The full range of uncertainty for the estimate of the amount of carbon that northern peatlands may accumulate from the start to the end of the current interglacial could be characterised by the uniform and clumped estimates. The former is equal to 665 PgC, and the latter is equal to 1258 PgC. However, our study shows that neither uniform interpretation nor clumped interpretation of the data on peatland extent is applicable everywhere, and hence the most likely range of uncertainty could be narrower than 665-1258 PgC. " – P5, L21-25 |
| 33. | Discussion:
General comments:
The discussion is very short. Please provide a more in-depth discussions of your methodological approach, e.g. show the benefits but also limitations of your model and compare your results of potential C accumulation with e.g. C accumulation during the Holocene. I suggest making several | The uncertainty of estimates is explained in the methods section (sub-section 2.3), because we apply an original method for characterizing the uncertainty of our estimates. | The discussion section was re-written based on the following logical scheme:
1. Discussing the novelty of the obtained estimate. P5. L26 – P6, L6
2. Warning about the uncertainty in the input data used in the study. – P6, L7-11
3. Arguing that despite all uncertainties, it is highly likely |

| | | | |
|---|---|---|---|
| | subchapters. One where you discuss the benefits and limitations of your model, including the uncertainty of your estimation. Another subchapter where you compare your results with previous studies (as you did in the results section) and a third subchapter where you discuss the implications of your results on the global C cycle (basically your actual discussion). | | that northern peatlands accumulate in the future more carbon than they store now. – P6, L12-26
 4. Discussing the range of validity of the obtained estimate with respect to anticipated climate change. – P6, L27 – P7, L6
 5. Discussing the implications of the obtained estimate to recovery of global carbon cycle – P7, L7-19 |
| 34. | Specific comments:
 P5, L.14: Change the first sentence into: "The potential for northern peatlands to store carbon were estimated for the first time: : :" | Yes, "potential for growth" and "limits to growth" have the same meaning, and "potential for growth" sound more positive. However, we would like to keep here a "connotation" to the paper "The limits to peat bog growth" by Clymo (1984) cited in the next phrase. | We did not make changes in response to this recommendation, but we can do this if necessary. |
| 35. | P5, L.15f: Change the following sentence into: "We adapted this methodology to global scale and additionally included geomorphological aspects of peat bog growth: : :" | We changed this sentence proceeding from general idea of this recommendation. | We adapted this methodology for use at the Earth system scale based on gridded data (Hengl et al., 2014) representing geomorphological aspects of peat bog growth. |
| 36. | P5, L.18 : Write "Our estimate: : :" instead of "Moreover, this: : :" | We deleted "Moreover". | "The estimate of potential carbon stocks, 875±125 PgC, corresponds to the present climate …"—P6, L27 |
| 37. | P5, L.18: Delete "somewhat" | Done | "and therefore assumes that the present climate is typical for the present interglacial period" – P6, L27-28 |
| 38. | P5, L.19: Change the following sentence into: "This assumption, however, might not be relevant for scenarios of dramatic changes in the Earth system that will take place | Done. | "This assumption, however, might not be relevant to the scenarios of dramatic changes in the Earth system, jeopardizing |

| | | | peatlands development." – P6, L28-29 |
|---|---|---|---|
| | if cumulative carbon: : :" | | |
| 39. | P5, L.20-21: Why 1000 PgC? It seems a bit arbitrary to me? Can you discuss this a bit more? | Done. | "The recent analysis of mitigation pathways compatible with global warming of 1.5°C above pre-industrial levels (Rogelj et al., 2018) shows that holding the global average temperature increase to well below 2°C is difficult but not impossible. To achieve this goal, cumulative $CO_2$ emissions from the start of 2018 until the time of net zero global emissions must be kept well below 1430 $GtCO_2$, (i.e., 390 PgC), that corresponds to 66[th] percentile of transient climate response to cumulative carbon emissions (Rogelj et al., 2018; Table 2.2). Since cumulative $CO_2$ emissions through to year 2017 are estimated at 610 PgC (Le Quéré et al., 2018), 1000 PgC of cumulative carbon emissions, the sum of historical (610 PgC) and the future cumulative emissions compatible with the global average temperature increase to below 2°C (390 PgC) could be considered as a threshold for defining the range of validity of the most expedient estimate of potential carbon stocks in northern peatlands." – P6, L29-P7, L5 |
| 40. | P5, L.21f: Change the following sentence into: "Nevertheless, if cumulative carbon emissions do not exceed 1000 PgC, the northern peatlands play an important role in global carbon cycle recovery" | Done. | "In brief, if cumulative carbon emissions do not exceed 1000 PgC, the northern peatlands play an important role in global carbon cycle recovery." – P7, L5-6 |

| 41. | P5, L.21: What happens to the peat C storage if carbon emissions exceed 1000 PgC? | According to Millar et al. (Nature Geoscience, 10: 741-747), 90% of CMIP5 models suggest that 468 PgC of cumulative carbon emissions after 2015 lead to warming by 1oC above 2010-2019 level under RCP2.6 scenario of radiative forcing. Hence, if cumulative carbon emissions exceed 1000 PgC (545 PgC from 1850 to 2015 plus 468 PgC after 2015), then one cannot exclude the risk of dramatic changes in climate leading to a massive destruction of northern peatlands. | We did not change the text in response to this comment, because peatland degradation goes beyond the scope of our study, but we could do this if necessary. |
|---|---|---|---|
| 42. | P5, L.26 : Replace "plain" with "other" | Done. | "In other words, the larger the perturbation of the Earth system, the lower the chances that the pre-industrial state will be restored in course of the current interglacial." – P7, L11-12 |
| 43. | P6, L.1-4: You should also discuss the conditions and timeframe under which such a scenario can happen. Is this only under ideal conditions? What about the limitations in the model? Also, if you make such a strong statement, there should be a better explanation of this Earth system model of intermediate complexity. | It is not a statement; it is rather a report about the numerical experiment that demonstrates the role of northern peatlands in global carbon cycle recovery and calls for further numerical experiments. That is why it is presented in discussion. The Earth system model of intermediate complexity, CLIMBER-2, is described in the cited paper. | We did not change the text in response to this comment, but we could do this if necessary. |
| 44. | P6, L.2: Maybe you can elaborate a bit more on figure 4. How does the orbital forcing affect peatland C uptake? | We did not consider the effect of orbital forcing on peatland C uptake in the reported numerical experiment. The purpose of this experiment was to show how additional long-term carbon sink provided by northern peatlands | We did not change the text in response to this comment, because the effect of orbital forcing goes beyond the scope of our study, but we could do this if necessary. |

| | | may affect the level of atmospheric CO2 concentration to which Earth system will return after the end of anthropogenic CO2 emissions. | |
|---|---|---|---|
| 45. | P6, L.2: "in relevant time frame" – Can you give a number, what a relevant time frame is? | The next reductions in northern summer insolation that may lead to glacial inception will occur 1500, 16000, and 53000 years after present. It is unlikely that atmospheric carbon dioxide concentration will return to the level typical for interglacial periods within next 1500 years. Hence, the next 5-15 thousand years is a relevant time frame for reducing the atmospheric carbon dioxide concentration to the level that is typical of interglacial periods. | "The northern peatlands are capable to remove in relevant time frame, that is, over the next 5-15 thousand years, the amount of carbon that ocean will not able to remove …" – P7, L17-18 |
| 46. | P6, L.3 replace "won0t" with "will not be able to" | Done. | "The northern peatlands are capable to remove in relevant time frame, that is, over the next 5-15 thousand years, the amount of carbon that ocean will not able to remove …" – P7, L17-18 |
| 47. | P6, L.7: What are limits to peatland growth? – Please discuss this in the discussion section | The limits to peatland growth are the peat depth values that cannot be exceeded in given climatic and geomorphologic conditions. | We did not change the text in response to this comment, because the phrase, "If there were no limits to their growth …", is written in a subjunctive mood: the growth of carbon stocks in any ecosystem is limited; everybody knows this. But we can change the text if necessary. |
| 48. | P6, L.10-16: This section is somewhat contradictorily in itself and compared to other parts of the manuscript. Why is the cumulative carbon removal associated with the | It seems that this impression results from using the logic of "subjunctive mood" here. First, we discuss what happens under an unrealistic assumption, "if | We did not change the text in response to this comment, because we think that this type of narrative is quite common. But |

| | natural development of peatland ecosystems limited? – Please discuss this in the discussion section | there were no limits …", then return to reality, "carbon removal associated with the natural development of peatland ecosystems is limited" and give the estimate of its potential magnitude. (See P7, L21-26) | we can change the text if necessary. |
|---|---|---|---|
| 49. | Supplement Please add a reference list for the supplement | Done. | We added the reference list to the Supplement. |
| 50. | S1.1 : Please rephrase the first sentence. | Done. | "The height of steady-state water table above the level of draining streams, $h$, satisfies the equation: "— S1.1 |

**Point-by-point reply to Reviewer #2 comments**

| No | Comment | Reply | Changes in the manuscript |
|---|---|---|---|
| 1. | General comments: In this manuscript, Alexandrov et al. present and discuss the estimates of northern peatlands carbon stocks using different approaches (conservative, non- and less-conservative approach). The procedure to calculate the total carbon content for the northern peatland areas have already been developed but in this study, authors have revised some values which they have estimated using the gridded soil dataset. The study has the potential to reduce the current uncertainties related to the limits of peatland carbon stocks and it is worth publishing. However, I find there are many sections which need to be strengthened, particularly, the methodology and result sections. | Done. | The "Methods" and "Results" are revised. |

| | | | |
|---|---|---|---|
| 2. | I also recommend them to divide the methods section into several parts under different sub-headings and include a brief explanation about the model in the beginning. | Done. | We subdivided "Methods" into "Model equations", "Input data", and "Uncertainty associated with peatlands distribution over a grid cell". |
| 3. | In the introduction and discussion sections, many arguments need to be referenced (see my comments below). More importantly, the authors have assumed that peatland distribution areas have not much been changed since the last 5000 years and the growth in the peat height was a major cause of carbon uptake in the northern areas. However, according to MacDonald et al. 2006 (see Figs. 1 and 3), around 30-40% peatlands were initiated after 5000 cal. B.P. which means that the increase in new peatland areas has also played a significant role in sequestering atmospheric carbon. How do they explain this assumption? | All comments were considered thoroughly and addressed in the revised manuscript. It seems to us that the assumption that in the late Holocene, the area of northern peatlands reached more than 70% of its maximum determined by geomorphological conditions does not contradict to the cited work, because MacDonald et al. (2006) write, "new peatland initiation was relatively modest in the late Holocene". If the new peatland initiation was relatively modest, the major part of the carbon sink resulted from the growth in peat depth. Taking into account considerable uncertainty over peatland area at any particular time of the past, we agree that it would be more correct to change "last 5000 years" to "late Holocene". | "Since the area of peatlands remained relatively stable in the late Holocene (Adams and Faure, 1998; MacDonald et al., 2006; Yu et al., 2010), the major part of the carbon sink provided by northern peatlands during this period could be attributed to the growth in peat depth, not to the growth of the area occupied by the northern peatlands." – P1, L26-29 |
| 4. | P1 L18: How did you define northern peatlands (> 40N or 45 N)? | Here we refer to the article of Loisel et al. (2017) and keep in mind the peatlands located north of 45 N. | "The recent compilations of peatland data (Loisel et al., 2014; Treat et al., 2019) largely confirm the conventional notion of the carbon (C) sink provided by northern peatlands, namely the peatlands distributed across the northern mid- and high-latitude regions located north of 45°N, since the Last Glacial Maximum (Loisel et al., 2017) " – P1, L20-23 |

| 5. | P1 L19: "The variations are explained by" : : : Which variations? | Here we keep in mind the variation in the carbon sink magnitude mentioned in the previous sentence (P1 L19). | "According to this notion, northern peatlands were providing a persistent but variable sink for atmospheric carbon (Yu, 2011). Variations in the sink magnitude are explained by changes in the rate of peatland expansion and in the rate of peat accumulation." – P1, L22-24 |
|---|---|---|---|
| 6. | P1 L22: "However, during the last 5000 years, the area of peatlands remained relatively stable: : :"

Peat basal ages are used as proxies to identify new peatland areas and expansion rate. From figures 1 and 3 in MacDonald et al. 2006, we can see that around 30-40% of the peatlands were initiated after 5000-year cal. B.P in northern areas. Therefore, I doubt whether the growth in the peat depth is the only major cause of carbon uptake in the past. | Here we cited the estimates of Bog/Swamp area given in the "Table 1. Reconstructed surface area of ecosystems" in the Adams and Faure (1998) article: 1.85 Mkm2 by 8000 BP, 2.35 Mkm2 by 5000 BP, 2.45 Mkm2 by present. The cumulative curve of 1516 radiocarbon dates of basal peat deposits shown at the at the Figure 3 in the article of MacDonald et al. (2006) also show that major part (70%) of the studied peatlands was initiated before 5000 BP. Moreover, MacDonald et al. (2006) wrote, "new peatland initiation was relatively modest in the late Holocene", that seemingly had the same meaning as the phrase "the area of peatlands remained relatively stable in the late Holocene", and led to conclusion that the growth in the peat depth was responsible for the major part of the carbon uptake in the late Holocene, whereas peatland expansion was responsible for the minor part of the carbon uptake in the late Holocene. (NB. We mean the part constituting more than 60% | "Since the area of peatlands remained relatively stable in the late Holocene (Adams and Faure, 1998; MacDonald et al., 2006; Yu et al., 2010), the major part of the carbon sink provided by northern peatlands during this period could be attributed to the growth in peat depth, not to the growth of the area occupied by the northern peatlands." – P1, L26-29 |

| | | | of the total under the words "major part".) | |
|---|---|---|---|---|
| 7. | P1 L25: "the northern peatlands may accumulate 864-2200 PgC : : :" This is a very high value, how did you calculate this range. From where did you find this information? What about the peatland distribution area and sink capacity, will they remain the same in the future? Studies indicated that many peatlands would lose their carbon sink capacity while some may enhance. | This range was calculated as follows. "The average rate of carbon accumulation associated with peat growth is estimated at 18-28 gC m-2 yr-1 (Yu, 2001). Northern peatlands occupy 2.4-4 million km2 (Yu, 2011). Hence, during the 20000 ($2*10^5$) years, the northern peatlands may accumulate from $(18*2.4*10^{12})*(2*10^4)=86.4*10^{16}=864*10^{15}$ gC to $(28*4*10^{12})*(2*10^4)=224*10^{16}=2240*10^{15}$ gC. This is an estimate of cumulative carbon uptake that could be provided by peatlands under the present peatland area and the present average carbon accumulation rate. Our research, in fact, is based on the hypothesis that average carbon accumulation rate must decline in the future. | "The average rate of carbon accumulation associated with peat growth is estimated at 18-28 gC m$^{-2}$ yr$^{-1}$ (Yu, 2011). This rate suggests that northern peatlands, occupying 2.4-4 million km$^2$ (Yu, 2011), may accumulate during the next 20,000 years the amount of carbon comparable to the expected cumulative anthropogenic carbon emissions corresponding to a 2.5°C warming (Raupach et al., 2014), namely from 864 (18 gC m$^{-2}$ yr$^{-1}$ ×2.4·10$^{12}$ m$^2$ × 2·10$^4$ yr) to 2240 (28 gC m$^{-2}$ yr$^{-1}$ × 4·10$^{12}$ m$^2$ × 2·10$^4$ yr) PgC." – P2, L1-5 |
| 8. | P2 L 1-15: Support your arguments with previously established knowledge. Include references. | Done. | "The process of reaching equilibrium can be conceptualized as follows, see also (Clymo, 1984; Alexandrov et., 2016)." |
| 9. | P2 L5: Define what a steady state is for your readers. | Done. | "Individual peatland development may lead to reduction of the carbon sink potential: the closer the peatland ecosystem is to its steady state, that is, to the equilibrium between organic matter production and decomposition, the lower is the carbon sink magnitude." |

| 10. | P2 L6: How did you estimate this range – see my previous comment. | This range was calculated as follows. "The average rate of carbon accumulation associated with peat growth is estimated at 18-28 gC m-2 yr-1 (Yu, 2001). Northern peatlands occupy 2.4-4 million km2 (Yu, 2011). Hence, during the 20000 (2*10^5) years, the northern peatlands may accumulate from (18*2.4*10^12)* (2*10^4)=86.4*10^16=864*10^15 gC to (28*4*10^12)*(2*10^4) =224*10^16=2240* 10^15 gC. | "The average rate of carbon accumulation associated with peat growth is estimated at 18-28 gC $m^{-2}$ $yr^{-1}$ (Yu, 2011). This rate suggests that northern peatlands, occupying 2.4-4 million $km^2$ (Yu, 2011), may accumulate during the next 20,000 years the amount of carbon comparable to the expected cumulative anthropogenic carbon emissions corresponding to a $2.5^{\circ}C$ warming (Raupach et al., 2014), namely from 864 (18 gC $m^{-2}$ $yr^{-1}$ ×2.4·$10^{12}$ $m^2$ × 2·$10^4$ yr) to 2240 (28 gC $m^{-2}$ $yr^{-1}$ × 4·$10^{12}$ $m^2$ × 2·$10^4$ yr) PgC." – P2, L1-5 |
|---|---|---|---|
| 11. | P2 L9: Remove this expression – "the so called" | Done. | "The plant litters do not enter the catotelm directly, but instead they first enter the upper layer of the peat deposit, the  acrotelm, that is not permanently saturated with water."—P2, L13-14 |
| 12. | P2 L11: "at least a small portion of the organic matter that enters the acrotelm always reaches to the catotelm : : :" Is this a plausible argument – do you think, acrotelm always passes organic matter in the catotelm? Even when peatland experiences continuous dry conditions? | Here we keep in mind an accumulating peatland, not a degrading peatland: "at least a small portion of the organic matter that enters the acrotelm always reaches the catotelm **in an accumulating** peatland". | We put more accent to the context of this phrase by adding the sentence, "This is, of course, not true in the case of a degrading peatland, but degrading peatlands do not fall within the scope of this study". – P2, L16-17 |
| 13. | P2 L 13-15: In which study, did you find this information? | This conceptual scheme summarizes and generalizes a number of studies, but the closest source is the article of Alexandrov, Brovkin, and Kleinen (Sci. Rep., 6, doi:10.1038/srep24784, 2016) | "The maximum height of the water table, and thus the potential peat depth, is determined by the amount of effective rainfall, drainage system density (the length of draining streams per unit area) and the hydraulic conductivity of peat and mineral materials below the peat |

| | | | (Alexandrov et al., 2016)." – P2, L20-22 |
|---|---|---|---|
| 14. | P2 L 16: Could you explain a bit about your model. What it does and other relevant information briefly and give more details in the methods section. | Done. | The section "Methods" is opened by explanation of model equations. |
| 15. | P2 L 22: "The gridded data on soil properties give the fraction of a grid cell covered by peatlands." Include reference. | Done. | "The gridded data on soil properties (Batjes, 2016) give the fraction of a grid cell covered by peatlands." – P4, L8 |
| 16. | P2 L27: How did you determine where to form a cluster in a grid cell? | The location of a peatland cluster in a grid cell does not affect the estimate of the potential amount of carbon that could be accumulated in the grid cell, therefore it is not determined. To avoid possible misunderstanding, we change "clustered distribution "to "non-uniform distribution": | "We address this uncertainty by giving three estimates of the potential amount of carbon that could be accumulated in northern peatlands: the uniform estimate, the clumped estimate and the conductivity-dependent estimate. The uniform estimate assumes a uniform distribution of peatlands over all grid cells ($f_{PW}= f_{P,obs}$; $f_{WP}=1$), the clumped estimate assumes a non-uniform distribution over all grid cells ($f_{PW}=0.75$; $f_{WP}=f_{P,obs}/0.75$) …" P4, L14-17 |
| 17. | P2: I think a paragraph needs to be added in the end which explains the purpose of your study | Done. | "The purpose of our study is to estimate the potential peat depth and carbon stocks over NH area north of 45°C and arrive to conclusion about the cumulative carbon removal associated with the natural development of northern peatlands by the end of the current interglacial."--P2, L23-25 |
| 18. | P3 L1: Methods Perhaps subheadings could be helpful to improve and clarify the structure of the methods. I also | Done. | We subdivided "Methods" into "Model equations", "Input data", and "Uncertainty associated with |

| | | | |
|---|---|---|---|
| | suggest you to add a model description section. | | peatlands distribution over a grid cell". |
| 19. | P3 L3: "The density of the draining system" – explain what it is? | Done. | "The maximum height of the water table, and thus the potential peat depth, is determined by the amount of effective rainfall, drainage system density (the length of draining streams per unit area) and the hydraulic conductivity of peat and mineral materials below the peat (Alexandrov et al., 2016)." – P2, L20-22 |
| 20. | P3 L4: "The impeded drainage model approach" – Give more details about this approach and model. What it is and where this approach has been used before? | The details of the impeded drainage model approach are given in the Supplement (https://www.biogeosciences-discuss.net/bg-2019-76/bg-2019-76-supplement.pdf). The basic idea of the approach is that the maximum peat depth is equal to the maximum height of water table above the level of draining system calculated using the impeded drainage model plus the maximum depth of acrotelm. | "To calculate the potential peat depth, we apply an equation derived (see Supplement) from the impeded drainage model used in our previous study (Alexandrov et al., 2016)." – P3, L3-5 |
| 21. | P3 eqn 1 - From where this equation comes from? Any previous applications? Please clarify. | This equation was derived from the equations of the impeded drainage model, see equations (S1-S17) in the Supplement (https://www.biogeosciences-discuss.net/bg-2019-76/bg-2019-76-supplement.pdf). | This part of the text is rewritten. |
| 22. | P3 L7-8: it is better to include the value of constants in the equation or under it. | Done | "where $d$ is the maximum depth of acrotelm, in m (set at 0.4 m)" –P3, L11
"where $c$ is the bulk carbon density of peat, in gC m$^{-3}$ (set at 58 KgC m$^{-3}$)" – P3, L14 |
| 23. | P3 L10: There are many peatlands in the southern latitude | Yes, we considered the peatlands located north of 45 N. | "The purpose of our study is to estimate the potential peat depth |

| | | | and carbon stocks over NH area north of 45°C and arrive to conclusion about the cumulative carbon removal associated with the natural development of northern peatlands by the end of the current interglacial."--P2, L23-25 |
|---|---|---|---|
| 24. | P3 L11: Include a brief write up about the SoilGrid dataset and what it contains. | We use only the depth to bedrock from this dataset and give reference to the paper where this data set is described in detail. | "The values of $g$ at the cells of 0.1°×0.1° geographic grid (Figure 1) were estimated from the data on depth to bedrock (Hengl et al., 2014)." |
| 25. | P3 L18: Did you check the recent study by Xu et al. 2018 where the authors have refined the global and regional estimates of peatland distribution area? How your dataset (WISE30sec) is different or better than Xu et al. 2018 (PEATMAP)? | The WISE30sec data set (Batjes, 2016) of soil properties is based on Harmonised World Soil Database (HWSD). The differences in the estimates of peatland area between HWSD and PEATMAP are reported in the Table 2 of the article published by Xu et al (2018): 1.327 vs 1,339 Mkm$^2$ for North America, 0. 879 vs 1.180 Mkm$^2$ for Asian Russia, 0.634 vs 0.528 Mkm$^2$ for Europe. It does not seem that these differences may dramatically affect our conclusion that it might be reasonable to agree that the estimate of 875±125 PgC, as obtained from two completely independent methods, is the most expedient estimate of the potential carbon stocks in northern peatlands. At the same time, we agree that it is important to trace the effect of input data updates. Therefore, we are going to publish the source code of the computer programs that were used in calculations. This source | "Analyzing the uncertainty in the data on present-day peatland extent goes beyond the scope of this study. Improving the accuracy of these data is a well known task tackled by ISRIC, the International Soil Reference and Information Centre, (Batjes, 2016; Hengl et al., 2014), and by networks of peatland scientists such as C-Peat (Treat et al., 2019) and PeatDataHub (Xu et al., 2018). Hence, it might be more important to update the estimates of potential carbon stocks on a regular basis to keep pace with improvements in the accuracy of the data on present-day peatland extent." – P6, L7-11 |

| | | code could be employed by anyone for updating our estimate in response to the updated information on peatland area. | |
|---|---|---|---|
| 26. | P3 L22: If you have the dataset then you can easily estimate how much area is occupied by northern peatlands. According to Xu et al., around 3.12 million km2 area is occupied by peatlands above 45N and Yu et al. 2010, used 4.0 million km2. | Yes. We corrected this phrase. | "These data allow us to estimate the values that $f_{P,obs}$ may take at the cells of the 0.1°×0.1° geographic grid (Figure 2) and the total area, 2.86 ×10$^6$ km$^2$, that peatlands occupy in the land north of 45ºN." – P3, L27-28 |
| 27. | P4 L: How accurate are these conservative and non-conservative estimates? From Table 1, one can see that both estimates fail to capture the observed peatland carbon density. In fact, in some cases, the conservative estimates are higher than the observed values. Based on this information, do you think we can rely on your modelled limits? | Both the conservative and non-conservative estimates are not the estimates of the present peat carbon density: they are estimates of the maximum peat carbon density that could be achieved in the future, under given climatic and geomorphological conditions. Therefore, they should not capture the observed peatland carbon density. The fact that the non-conservative estimates are significantly higher than the observed carbon densities allows the following interpretation: the sites listed in the Table 1 are far from equilibrium and could accumulate a large amount of carbon by the end of the current interglacial. As to the conservative estimates, which are lower than the actual peat carbon density at the sites that fall within the grid cells where fP,obs is less than 20%, this suggests that there are few large peatlands, not many small peatlands in these grid cells. The question about reliability of the | This part of the text was completely re-written, see P5, L1-11 |

| | | estimates of modelled limits to peat growth is a difficult question. Most projections of the future are reliable only in theoretical sense, under some assumptions. The estimate of the carbon stock that could be accumulated by northern peatlands during the current interglacial is probably not less reliable than the estimates of the carbon stock that northern peatlands accumulated by present time. | |
|---|---|---|---|
| 28. | P4 Results: This looks like a part of the discussion. I recommend you to explain your results and what you see in your figures before comparing them with the previously established knowledge. For example, you can highlight how much peatland carbon stocks you have estimated in the European, Russian and N. American regions, which areas are rich in carbon within in these regions, what are the maximum and minimum peat depths, why you used a constant bulk density value etc. | Done. | The section "Results" is completely re-written, see P5, L14-L25 |
| 29. | P4 L27: You can also include the eqn used by Gorham 1991- Cpeat = Pi (Ai x Di x BDi x CCi) | We did not find this equation in the cited Gorham's paper. Therefore, we supposed that it might be better to explain the Gorham's version of the peat-volume approach through an indirect quotation of his words. Here is the direct quotation, "… we can then estimate readily the total carbon in the dry mass of boreal and subarctic peat, subtracting the mined area, as | No changes made in the manuscript in response to this comment. |

| | | (3.42 x 10^12 m^2) x (2.3 m) x (112 x 10^3 g/m^3) x (0.517) = 455 x 10^15 g, or 455 petagrams (Pg)." | |
|---|---|---|---|
| 30. | P4 L29: 112 x 10^3 g m-3 - Change it to 112 kg m-3 | Done | "mean bulk density of peat (112 Kg m$^{-3}$))," – P6, L16 |
| 31. | P5 L4: Explain what time history approach is. | Since it is difficult to explain this approach in few words, we suppose that it might be better to give a reference to the publication, where this approach is explained in detail. | "estimate of actual carbon stocks, 547±74 PgC, based on the time history approach (Yu et al., 2010)" – P6, L19 |
| 32. | P5 L15: There are other methodologies which have been developed to estimate total carbon stocks (see Yu et al. 2012). How your approach is different or better than these methodologies and what are its limitations? | These methodologies are to estimate present carbon stocks. We estimate the carbon stocks that could be achieved in the future. Our approach for estimating the future carbon stocks is similar to peat volume approach, but the estimate of the mean peat depth in a given region is replaced by the estimate of the maximum mean peat depth that could be achieved in the given region. | "The uniform estimate is also higher than the Yu's (2011) estimate of actual carbon stocks, 547±74 PgC, based on the time history approach (Yu et al., 2010), suggesting that northern peatlands in total would accumulate in the future more carbon than they store now." – P6, L17-20 |
| 33. | P5 L18-22: No references. | References were inserted. | This part of the text was completely re-written, see P6, L28 – P7, L4 |
| 34. | P5 L16: "We adapted this methodology for use at the global scale : : :" Global or regional because you have considered only the northern peatlands? | It may be more correct to say, that we adapted this methodology for use in the studies of the Earth climate system, as we found that northern peatlands are the important element of the Earth climate system affecting the length of the current interglacial. | "We adapted this methodology for use at the Earth system scale based on gridded data (Hengl et al., 2014) representing geomorphological aspects of peat bog growth." – P5,L29- P6, L2 |
| 35. | P6 L7: "If there were no limits to their growth: : :" In the introduction, you have mentioned that peatlands can reach to steady state and do not | If there would be a map of carbon stocks in peatlands, the comparison of this map to the map displayed at the Fig.3 (the less-conservative estimate of the | No changes made in the manuscript in response to this comment. |

| | | grow or accumulate carbon after that. Do your analysis shows in which regions peatlands have already reached to steady state? | potential carbon stocks) would show in which regions peatlands have already reached to steady state. At the moment, only the grid cells where $p_{C,max}/(A*f_{P,obs}) \le 45$ KgC m-2 could be categorized as the grid cells where peatlands already reached the steady state. | |
|---|---|---|---|---|

[revised manuscript text omitted]

---

## Author Response (AR2)

**The limits to northern peatland carbon stocks**

by G. A. Alexandrov, V. A. Brovkin, T. Kleinen, and Z. Yu

*Biogeosciences Discuss*.,

https://www.biogeosciences-discuss.net/bg-2019-76/#discussion

Point-by-point reply to comments

20    All comments were considered thoroughly and were addressed in the revised manuscript. The help of reviewers is acknowledged in the revised manuscript.

Besides recommended revisions, some language editing was done to improve readability of the manuscript.

**Point-by-point reply to Editor's comments**

| No | Comment | Reply | Changes in the manuscript |
|---|---|---|---|
| 1. | I have now received the reviewer opinion on your revised manuscript. Please implement those demanded changes to the manuscript as outlined by the reviewer. I think your manuscript will greatly profit from these additional improvements. | The manuscript was changed as recommended. | These changes were made at the following pages of the marked-up manuscript version starting from page 8 of this file.

P7 (title and abstract); P11 line 18; P9 lines 26-28, at P10 lines 12-13, at P11 line 32-P12 line 4; P12 lines 15-20; P15 lines 10 – 16. |

**Point-by-point reply to Reviewer #3 comments**

| No | Comment | Reply | Changes in the manuscript |
|---|---|---|---|
| 1. | Version 2 of the manuscript has been significantly improved by the authors employing reviewers' comments and suggestions. The introduction and the method sections are easier to understand and highlight objectives of the study as well as limits related to the methodology employed and hypothesis made to estimate the ability of northern peatlands to reduce atmospheric $CO_2$ content in | Thanks for this positive assessment of the revised manuscript. | N/A |

| | | | |
|---|---|---|---|
| | the future. This study fuels the discussion on the interactions of northern peatlands with climate and provide a quantification of one of the interactions which is the incorporation and accumulation of atmospheric $CO_2$ into peatland soil. | | |
| 2. | Nevertheless, it is a shame that the title does not reflect the whole content and significance of this work. The authors insist on having a reminder in the title of the work by Clymo (1984) "The limits to peat bog growth" in order to highlight the fact that "the cumulative amount of carbon that northern peatlands could remove from the atmosphere is limited by the geomorphological conditions in present climate." (response to comment 3 of reviewer 1 in point by point reply to all comments document) while the main hypothesis used to compute the estimations is that northern peatland surface area does not significantly change over a period of thousands of years. This hypothesis is fair to take for the present work but it is a strong constrain on the results. Therefore, I think that the authors should propose | This comment sounds very convincing. It would be logical to change the title (and the abstract) to shift the focus to the capacity of northern to remove a large amount of carbon from the atmosphere at a long-term scale. | P7: The title of the manuscript was changed to "The capacity of northern peatlands for long-term carbon sequestration" The abstract was revised to fit the changed title. |

| | | | |
|---|---|---|---|
| | a title that will be appealing for potential readers and reflex in a more suitable manner the content or the topic of the manuscript. I understand that the authors do not like the word "potential" as suggested by reviewer 1 comment 3 "the potential of northern peatlands for carbon sequestration" fortunately for us the English language is full of words and there are plenty of synonyms for "potential or limits": ability, capacity, faculty, potency … that could be used here. | | |
| 3. | Fews minor errors in the text of manuscript version 2:

P5 line1" The use of this approach to addressing uncertainty" choose the either forms "this approach to address" or "this approach addressing"

P7 line 18 "that ocean will not able to remove" it is missing the verb to be, please modify to "that ocean will not be able to remove" | We corrected the errors. | P11 line 18: "The use of this approach to addressing uncertainty" was changed to "This approach to address uncertainty".

The phrase "that ocean will not able to remove" falls within the text removed in response to the Comment #5. |
| 4. | I do regret that there are only so little description and comments on figures and table content. It feels like there are only there as nice illustration | Done | The descriptive statistics of the data sets illustrated by Figures 1-3 was added at P9 lines 26-28, at P10 lines 12- |

| | | | |
|---|---|---|---|
| | where the authors could have talk more about geographical variability from a region to another as suggested by reviewer2 comment28 of the point by point reply to all comments documents. For example, the authors response to reviewer 2 comment 27 is very interesting on the accuracy of both estimates presented in Table 1. I think this could be added to the manuscript into the results section. It shows how interesting is your estimation approach and values, are at lower scale and the variability of the estimate over multiple sites. | | 13, and at P11 line 32 – P12 line 4.

The results of comparison estimated and observed carbon densities presented in the Table 1 were added at P12 lines 15-20. |
| 5. | Regarding figure 4 and simulation results from CLIMBER-2 Earth system model authors argued to review 1 comment 43 that it "demonstrates the role of northern peatlands in global carbon cycle recovery and calls for further numerical experiments." In the manuscript model description is summarized in one sentence (supplementary document page 6 first sentence under section S4 Numerical experiments on the CLIMBER-2 ) and simulation set up and conditions are not described. In addition, the | Done | These parts of the text were removed. P15 lines 10-16. |

| | | |
|---|---|---|
| goal of the present work is to provide an estimate of the capability of northern peatland to store carbon therefore these simulations are beyond the scope of the paper. Since the authors estimations show an increase in peatland carbon storage in the future, they already demonstrate the role of peatland in the global carbon cycle rather than the simulation results that show that if you add another carbon sink ecosystem that is independent to the first one (the ocean) you do sequester more carbon from the atmosphere which is obvious. And regarding the effect of orbital forcing, in the response to review 1 comment 44, the authors recognized that is beyond the scope of the present study.

Therefore, for all the above reasons, you should remove figure 4 and the final paragraph line 13 to 19 page 7 of the discussion section from the manuscript and S4 from supplementary documentation. | | |

[revised manuscript text omitted]